# Consecutive multimaterial printing of biomimetic ionic hydrogel power sources with high flexibility and stretchability

Pei He[1,2], Junyu Yue[1,2], Zhennan Qiu[1,2], Zijie Meng ®[1,2,3], Jiankang He ®[1,2] ✉ & Dichen Li[1,2]

Electric eel is an excellent example to harness ion-concentration gradients for sustainable power generation. However, current strategies to create electric-eel-inspired power sources commonly involve manual stacking of multiple salinity-gradient power source units, resulting in low efficiency, unstable contact, and poor flexibility. Here we propose a consecutive multimaterial printing strategy to efficiently fabricate biomimetic ionic hydrogel power sources with a maximum stretchability of 137%. The consecutively-printed ionic hydrogel power source filaments showed seamless bonding interface and can maintain stable voltage outputs for 1000 stretching cycles at 100% strain. With arrayed multi-channel printhead, power sources with a maximum voltage of 208 V can be automatically printed and assembled in parallel within 30 min. The as-printed flexible power source filaments can be woven into a wristband to power a digital wristwatch. The presented strategy provides a tool to efficiently produce electric-eel-inspired ionic hydrogel power sources with great stretchability for various flexible power source applications.

Salinity-gradient energy conservation has recently gained incremental attention as a promising clean and renewable energy source relying on ion transportation[1–3]. As an excellent example of salinity-gradient energy generators, electric eels in nature have unique electric organs (EOs), which consist of highly-aligned and serially-interconnected electrocytes to produce high electrical discharge up to 800 V via transmembrane ionic transportation[4]. Such unique capability in EOs has inspired the biomimetic design of high-efficient power generation systems based on salinity-gradient-induced reverse electrodialysis (RED)[5,6]. For instance, Wang et al. proposed to fabricate artificial power source units with microchannels separated by alternate cation- and anion-exchange membranes as the wall of microchannel network via in-situ self-assembling of nanoparticles with hydroxyl and amine groups to mimic the configuration of electrocytes in EOs[7]. When the cations and anions in the microchannels filled with concentrated solutions were transported across the nanomembranes to the

microchannels with low ion-concentration solutions, the ionic movement inside a single artificial unit can produce a diffusion potential of 138 mV similar to an electrocyte, which can be improved to 1 V by stacking 20 artificial units in series. Nevertheless, the complicated fabrication strategy including photolithography, microfabrication[8,9] and in situ self-assembly of the nanoparticles largely limited the number of serially-connected units for the high voltage output of the resultant power generation system[6,10,11].

To improve the power capacity of RED-based electric generators, it is crucial to develop high-efficient manufacturing techniques to fabricate power generation systems with an expanding number of biomimetic power source units. Schroeder et al. employed a surface printing strategy to sequentially fabricate high- and low-concentration ionic hydrogel particles on one substrate as well as the cation- and anion-exchange hydrogel particles on another substrate. The resultant two substrates were manually stacked together to connect the two

[1]State Key Laboratory for Manufacturing System Engineering, Xi'an Jiaotong University, Xi'an 710049, China. [2]NMPA Key Laboratory for Research and Evaluation of Additive Manufacturing Medical Devices, Xi'an Jiaotong University, Xi'an, Shaanxi 710049, China. [3]Frontier Institute of Science and Technology, Xi'an Jiaotong University, Xi'an 710049, China. ✉e-mail: jiankanghe@mail.xjtu.edu.cn

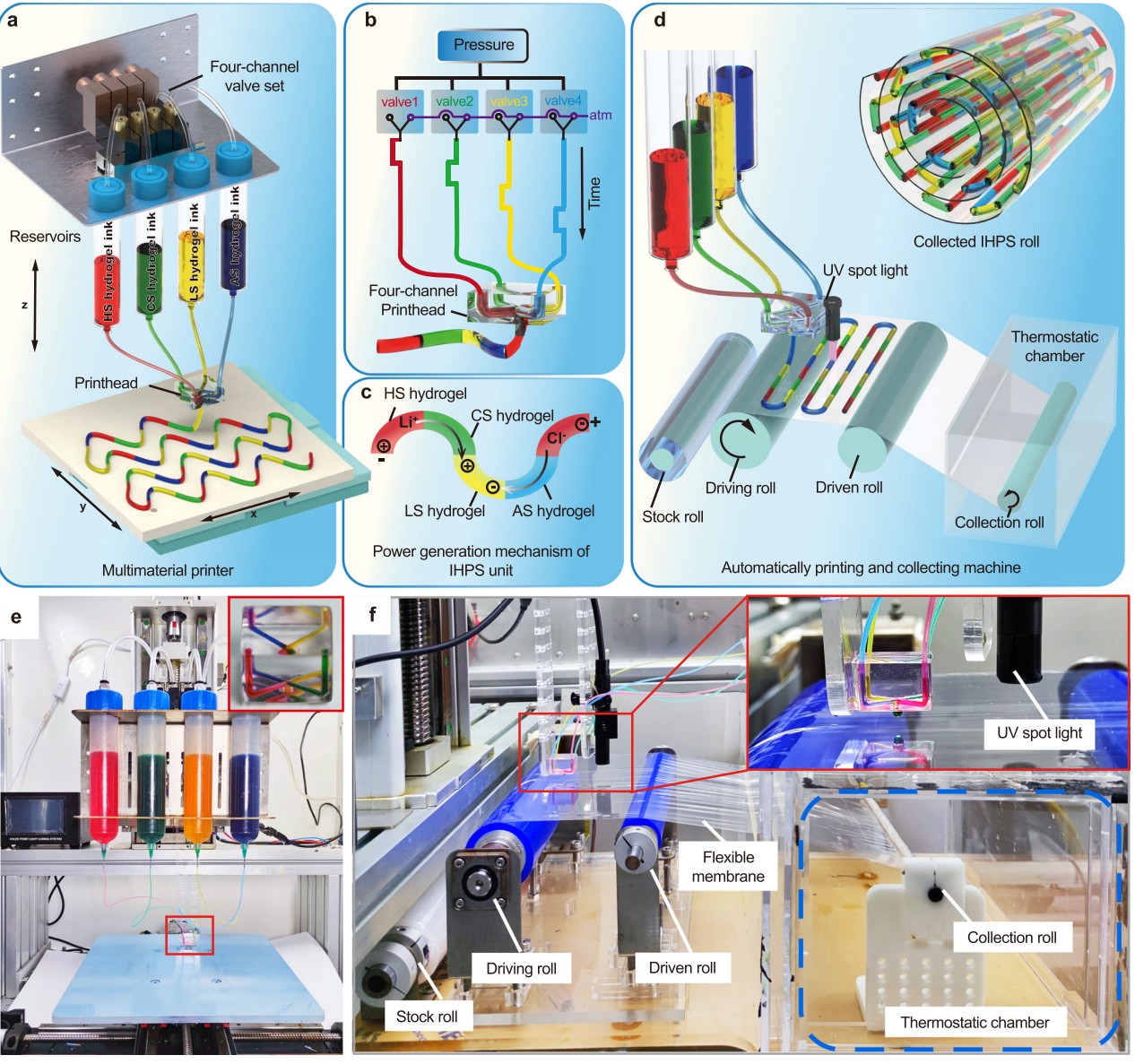

**Fig. 1 | Schematics of consecutive multimaterial printing process for biomimetic ionic hydrogel power source (IHPS). a** Schematics of the consecutive multimaterial printing setup. **b** Programmable-controlled four-channel valve set for the dynamic switching of HS ink (marked with red), CS ink (marked with green), LS ink (marked with yellow), and AS ink (marked with blue) inside a four-channel printhead. **c** Power generation mechanism of the consecutively-printed IHPS unit. **d** Schematics of the automated consecutive multimaterial printing system with collection modules. The inset showcases the automatically-collected IHPS roll. **e, f** Photograph of the house-made multimaterial printing configuration integrated with a roll-to-roll printing and collection module.

kinds of ionic hydrogels via the ion-exchange hydrogels, which formed a biomimetic power source unit[12]. This technique enabled the fabrication of a salinity-gradient-induced hydrogel power generator with 612 units in series and a maximum voltage output of 110 V. To improve the fabrication efficiency, we previously developed a microfluidics-based perfusion strategy to fabricate ionic and transmembrane hydrogel particles with four kinds of material compositions on two flexible substrates, which can realize the formation of 800 particles within 1.53 min[13]. The two substrates were manually stacked and tightly sealed together via negative pressure to obtain a flexible ionic hydrogel power source with great flexibility. To improve the current output, ionic hydrogel power sources with shortened ionic transport pathway were constructed by infiltrating four hydrogel precursor solutions into thin paper films (hundreds of microns thick) and then stacking them sequentially. The resultant paper-based ionic power source exhibited good flexibility and a higher power density of 1.80 W m$^{-2}$ due to the

lower transmembrane resistance of hydrogel-penetrated films in comparison with hydrogel particles[14]. However, these existing strategies commonly relied on the manual assembly of two or multiple layers of different hydrogel components, which highly affected the interface contact stability of the resultant ionic hydrogel power sources, especially when working in considerable flexibility and stretchability conditions.

Here we proposed a consecutive multimaterial printing strategy to automatically fabricate biomimetic ionic hydrogel power sources (IHPS) with high flexibility and stretchability by harnessing ion-concentration gradients. Four types of ionic hydrogel inks with similar viscosity and rheological properties were specifically developed for consecutive multimaterial printing including high-salinity (HS), cation-selective (CS), low-salinity (LS) and anion-selective (AS) precursor solutions (Fig. 1a). Pneumatic pressure was employed to sequentially and periodically extrude four kinds of ionic hydrogel inks via a

microfluidic printhead with four-channel inlets and one outlet as shown in Fig. 1b. The extrusion sequence was programmatically controlled by the electromagnetic valves, which enabled the rapid switching of pneumatic pressure as well as the consecutive printing of four components in a pre-defined order of HS, CS, LS, and AS. The printed multimaterial filaments were collected on a moving XY stage and photo-crosslinked via ultraviolet exposure. The resultant constructs allowed the flow of lithium ions ($Li^+$) and chloride ions ($Cl^-$) from the HS hydrogel to LS hydrogel via CS hydrogel and AS hydrogel respectively as shown in Fig. 1c, which formed bionic IHPS units similar to native electrocytes.

To facilitate large-area printing and automatic collection of the resultant IHPS for high voltage output and flexibility, a roll-to-roll collection module was specifically designed and integrated with the consecutive multimaterial printing system as shown in Fig. 1d. The collection module contains four rolls including the stock roll with flexible substrate, the driving roll, the supporting roll, and the collection roll. The flexible substrate was continuously fed by the rotation of the driving roll to form a flat printing zone between the driving and supporting rolls where the IHPS was consecutively printed and photo-crosslinked with an ultraviolet (UV) spot lamp. The printed IHPS along with the flexible substrate was automatically rolled into a cylindrical construct by the reverse rotation of the collection roll inside a temperature-controlled chamber. Figure 1e, f show the house-made multimaterial printing system for the consecutive fabrication of biomimetic IHPS with a large unit number, which mainly includes a pneumatic generator, four-channel valve-based switching module, microfluidic printhead, the roll-to-roll feeding and flexible collection module.

## Results

### Optimization of the multimaterial printing process

The rheological properties of the ionic hydrogel inks are crucial for the consecutive multimaterial printing process, which requires different ink having similar viscosity, shear-thinning behavior, and shape-holding capability[15,16]. The previously-developed four types of IHPS precursor solutions exhibited low viscosity of *c.a.* 2.5 mPa s$^{-1}$ and can not be directly used for the consecutive extrusion of multimaterial filaments (Table S1). In this study, hydroxyethyl cellulose (HEC) and polyethylene oxide (PEO) were incorporated into the previously-established basic formula after replacing NaCl (2.5 mol $L^{-1}$) in HS with a higher concentration of LiCl (6.0 mol $L^{-1}$) for enhanced printability and electrical potential[13]. Specifically, HEC with different concentrations of 1.0%–1.3% (w/v%) was added as a rheology modifier to regulate the rheological and shape-holding behaviors of the inks while 0.05% PEO was supplemented for improved stretchability. Figure 2a shows the viscosities of the original HS precursor solution as well as the modified HS inks with different HEC content. It can be seen that the addition of HEC and PEO endowed the resultant HS inks obvious shear-thinning behavior and significantly improved the ink viscosity in comparison with the original HS precursor solution. In addition, we found that the printed HS fibers were prone to spread out on the substrate at an HEC content smaller than 1.2% while the filamentary shape can be well maintained at an HEC content of 1.3% (Supplementary Fig. 1). However, further increasing the concentration of the non-conductive rheology modifier HEC resulted in increased internal resistance of the hydrogel matrix, which will hinder the power output of the resultant IHPS. As a result, the concentration of HEC in the HS ink was optimized at 1.3% to minimize internal resistance while ensuring good printability. The same strategy was further applied to optimize the HEC concentration in the original LS, CS, and AS precursor solutions. The results indicated that when the PEO content was kept at 0.05% and the concentration of HEC in the LS, CS, and AS inks was 1.52%, 1.35%, and 1.82% respectively, each optimized ink showed a similar viscosity curve to the HS inks (Fig. 2b). The storage modulus (G') and loss modulus (G'') of the

optimized four types of IHPS inks are shown as a function of the angular frequency in Fig. 2c. The G'' significantly surpassed G' at high angular frequencies, which is a favorable rheological property for extrusion-based 3D printing[17]. In contrast, G'' slightly exceeded G' in the low-frequency range below 10 rad/s, indicating that the inks exhibited a semi-gel state under a low shear rate. These properties are very important for the rapid extrusion and switching of the developed four inks inside the multimaterial printhead, and simultaneously maintain the shape fidelity of the hydrogel filaments after extruded from the printhead. The four types of hydrogels with optimized contents of rheology modifiers were further characterized by FTIR and Raman spectroscopy. The peaks observed at 3200–3530 cm$^{-1}$ in the FTIR spectra of four types of hydrogels belong to the N–H stretching of polyacrylamide, while the peaks around 1652 cm$^{-1}$ are the C = O characteristic absorption peak of it (Supplementary Fig. 2). The broad band at 2450–3500 cm$^{-1}$ of CS hydrogel was attributed to the N–H stretching, C–H stretching of polyacrylamide, and O–H stretching of the cation-selective reagent, respectively. A relatively large number of hydrogen bonds were formed in the Raman spectra of HS hydrogel, due to the addition of a large amount of LiCl[18], resulting in good anti-freezing property.

Another important factor to realize successful consecutive printing of multiple hydrogel components is the intersection angle between the inlet channels and the outlet channel in the microfluidic printhead, which determines the switching length of the IHPS[19]. To optimize the intersection angle ($\alpha$), the printhead was firstly designed with two inlets and one outlet according to previously-established literature[20] with $\alpha$ varied in the range from 90° to 150° (Fig. 2d). The channel diameter was fixed at 1.5 mm and was fabricated by stereo-lithography (Yichangtai Intelligent Machine Co., Ltd., China). As shown in Fig. 2d, e, when the intersection angle gradually decreased from 150° to 120°, the switching length was significantly reduced from 5.31 ± 0.19 mm to 2.12 ± 0.13 mm. Further decrease in the intersection angle from 120° to 90° showed little effect on the final switching length, which maintained at a relatively stable level of 2 mm (Fig. 2e). The shorter switching length indicates the smaller interface area between two inks, which could potentially avoid material mixing during the multimaterial switching process. Therefore, for the four-channel microfluidic printhead, the branched channels with four separate vertical inlets were evenly designed in three dimensional and the intersection angle between the branched channels and the outlet channel was uniformly fixed at 90° as shown in Fig. 2f.

To realize the controllable multimaterial printing of IHPS, the effect of the air pressure and moving speed on the morphology and width of the IHPS was investigated using the developed multimaterial printhead. Figure 2g shows the printed IHPS unit when the air pressure varied from 80 kPa to 180 kPa while the moving speed, switching frequency, and nozzle-to-collect distance were fixed at 4.0 mm s$^{-1}$, 1/4 Hz, and 5.0 mm respectively. In all cases, the IHPS unit can be consecutively printed with four types of hydrogel filaments seamlessly connected in sequence. At a lower extrusion pressure, the filament morphology was highly affected by the pressure switching process, which caused a smaller filament width at the interface region. The average filament width gradually increased from 1.97 ± 0.55 mm to 3.70 ± 0.89 mm (Fig. 2h) with extrusion pressure increasing from 80 kPa to 180 kPa. The uniformity of filament morphology was improved by increasing the extrusion pressure from 80 kPa to 140 kPa and reached a maximum uniformity of 0.89 ± 0.04 at 140 kPa, which decreased to 0.77 ± 0.09 as the air pressure further increased above 160 kPa.

To further improve the uniformity of the printed hydrogel filament, the stage moving speed was changed from 2.0 mm s$^{-1}$ to 5.0 mm s$^{-1}$ when the air pressure, switching frequency, and nozzle-to-collect distance were fixed at 140 kPa, 1/4 Hz, and 5.0 mm respectively. As shown in Fig. 2i, j, as the moving speed decreased from 5.0 mm s$^{-1}$ to

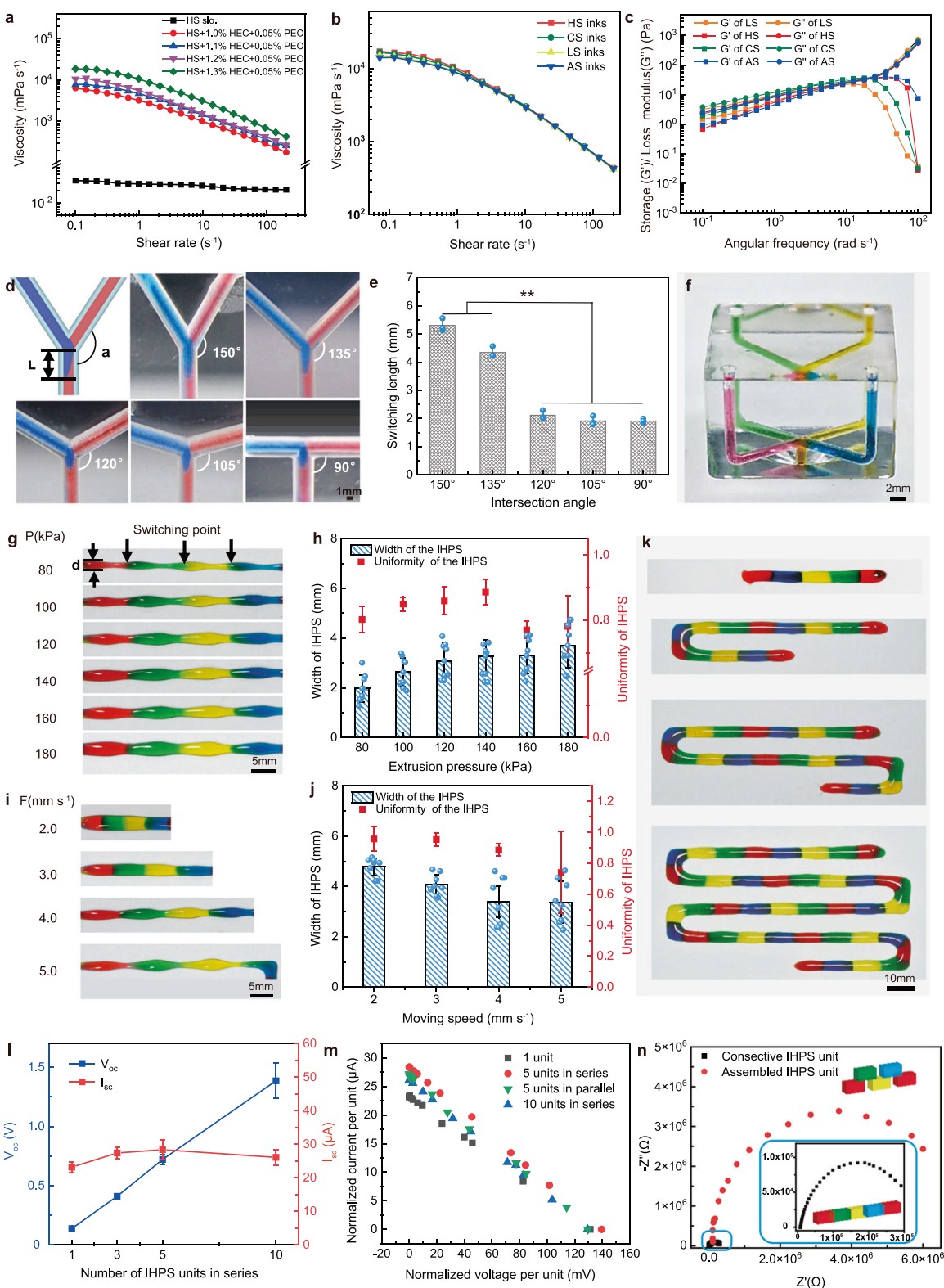

2.0 mm s$^{-1}$, the filament size increased from $3.40 \pm 0.81$ mm to $4.78 \pm 0.35$ mm while the uniformity was found to significantly increase from $0.74 \pm 0.26$ to $0.95 \pm 0.04$. There is no significant difference between the uniformity of the filaments printed at the moving speed of 2.0 mm s$^{-1}$ and 3.0 mm s$^{-1}$. The stage moving speed and air pressure were chosen as 3.0 mm s$^{-1}$ and 140 kPa for the multimaterial printing of IHPS with a switching frequency of 1/4 Hz, which can ensure the stable and controllable multimaterial printing of consecutive IHPS with a unit

length of $48.03 \pm 0.13$ mm and a uniform filament width of $4.07 \pm 0.39$ mm. Figure 2k displays the consecutively-printed IHPS with different unit numbers, exhibiting uniform macroscopic morphologies and filament size.

## Electrical properties of the consecutively-printed IHPS

Figure 2l shows the open-circuit voltage ($V_{oc}$) and short-circuit current ($I_{sc}$) of the consecutively-printed IHPS with a unit number of

**Fig. 2 | Optimization of consecutive multimaterial printing strategy for the fabrication of biomimetic ionic hydrogel power sources (IHPS). a** Effect of different HEC and PEO contents on the apparent viscosity of the resultant HS inks. **b** Apparent viscosity of optimized IHPS inks at the shear rate ranging from 0.01 to 200 $s^{-1}$. **c** Storage modulus and loss modulus as a function of the angular frequency of the optimized IHPS inks. **d** Effect of different inlet-outlet intersection angles on the switching interface and (**e**) switching length (mean ± s.d., $n = 3$) between two distinct inks inside the printhead. Statistical comparisons were conducted using one-way Analysis of Variance with Tukey's tests to determine statistical significance among multiple groups. The levels of significance were established at **$P < 0.01$. **f** The photograph of the developed four-channel multimaterial printhead. **g** The photograph, (**h**) width (mean ± s.d., $n = 8$) and uniformity (mean ± s.d., $n = 4$) of consecutively-printed IHPS filaments when the air pressure varied from 80 kPa to

180 kPa while the moving speed, switching frequency, and nozzle-to-collect distance were fixed at 4.0 mm $s^{-1}$, 1/4 Hz and 5.0 mm respectively. **i** The photograph, (**j**) width (mean ± s.d., $n = 8$) and uniformity (mean ± s.d., $n = 4$) of IHPS filaments when the moving speed varied from 2.0 mm $s^{-1}$ to 5.0 mm $s^{-1}$ while the air pressure, switching frequency, and nozzle-to-collect distance were fixed at 140 kPa, 1/4 Hz and 5.0 mm respectively. **k** Photograph, (**l**) $V_{oc}$ and $I_{sc}$ of the printed IHPS filaments with 1, 3, 5, and 10 units (mean ± s.d., $n = 5$). **m** Normalized current−voltage relations of the printed IHPS with various unit numbers connected in series or parallel. Voltage is normalized by the number of units in series, while current is normalized by the number of units in parallel. **n** The Nyquist plots of consecutively printed and manually assembled IHPS unit were compared. Source data are provided as a Source Data file.

1, 3, 5, and 10. The $V_{oc}$ of a single IHPS unit was 137.89 ± 18.10 mV, which increased with a larger concentration gradient between HS and LS (Supplementary Fig. 3). When the unit number increased from 3, 5, to 10, the $V_{oc}$ was linearly improved from 0.41 ± 0.02 V to 0.72 ± 0.04 V, and finally to 1.39 ± 0.15 V. However, the average $I_{sc}$ of the corresponding IHPS remained basically unchanged at c.a. 27 μA, which might be attributed to the linearly-increased resistance of the resultant IHPS.

As shown in Fig. 2m, when the IHPS containing 5 and 10 units in series or parallel were connected with different resistors with the value ranging from 1 kΩ to 1 MΩ, the normalized current and voltage curves exhibited a similar and linear profile to the single IHPS unit regardless of the sequentially- or parallelly-connected configuration. The IV curve of the IHPS was further characterized by using the linear sweep voltammetry. The $V_{oc}$ and $I_{sc}$ of IHPS unit read from the intercepts on the voltage and current axes were 134.81 ± 6.62 mV and 25.98 ± 2.31 μA (Supplementary Fig. 4)[21], respectively, which was consistent with the results of the normalized current and voltage curves. When the unit number increased to 5, the corresponding $V_{oc}$ linearly increased to 670.43 ± 26.49 mV while the $I_{sc}$ remained relatively stable at 26.01 ± 1.32 μA. The aforementioned phenomena serve as evidence that the current and power output of consecutively-printed IHPS is directly proportional to the number of units, which were analogous to the electrocytes in EOs of native eels and consistent with previous findings in the manually-stacked IHPS[12].

The electrochemical property of the as-printed IHPS unit was further characterized, while the IHPS unit with two layers fabricated by microfluidics-based perfusing was used as the control[13]. The Nyquist plots of both IHPS units are composed of dome shapes with the diameter corresponding to the resistance of charge transfer. As shown in Fig. 2n, the semicircle of the Nyquist plots of the consecutively-printed power source unit was much smaller than that of the manually-stacked power source unit, indicating reduced charge transfer resistance at the interface region (Supplementary Fig. 5)[22,23]. This might be mainly attributed to the tightly-bonded and seamless interface between the adjacent hydrogel compositions achieved by the consecutively multimaterial printing technique. The consecutively-printed IHPS was packed with flexible substrate (polyethylene film with a thickness of 200 μm) aiming to avoid water loss and simultaneously ensuring electrical isolation between the printed ionic hydrogel filaments to prevent short-circuiting even under the large deformation conditions. The film-packed IHPS maintained a stable water content across a broad range of temperature and humidity (Supplementary Fig. 6), which is crucial for long-term stable electric discharge.

### Stretchability and electrical stability of the IHPS

The consecutive multimaterial printing process also endowed the resultant IHPS with excellent stretchability. Figure 3a displays the typical stress-strain curves of the consecutively-printed IHPS unit containing four types of hydrogel inks and the hydrogel filaments derived from single ink, respectively. It was found that all the printed hydrogel filaments presented a linear increase in stress as the strain

increased. The CS hydrogel showed the highest elongation of 200.05% ± 14.38%, and that of HS, LS, and AS hydrogel was 77.57% ± 9.64%, 154.13% ± 31.56%, and 103.21% ± 41.62%, respectively. The corresponding modulus of HS, LS CS, and AS hydrogel was 162.65 ± 19.66 Pa, 48.31 ± 12.33 Pa, 105.52 ± 18.77 Pa, and 122.59 ± 11.6 Pa, respectively (Fig. 3b). When the four types of hydrogels were consecutively printed to form a seamless IHPS unit, it exhibited compromised flexibility with an elongation of 137.47 ± 26.97% and a tensile modulus of 96.58 ± 6.83 Pa. Since the moduli varied from one hydrogel portion to another, the consecutively printed IHPS unit exhibited nonuniform stretching with the largest elongation at the LS hydrogel phase due to its lowest modulus (Fig. 3c). Moreover, when the strain further increased to c.a. 137%, the LS hydrogel fractured first since it has the lowest break strength of 6.27 ± 0.32 kPa in the IHPS unit. (Supplementary Fig. 7). No fracture was observed at the interface regions, verifying reliable bonding among different ionic hydrogel phases[24]. Besides, cyclic tensile and compression tests were conducted on the hydrogel samples. The tensile stress-strain curve of the IHPS unit showed a high degree of overlap over strains ranging from 20% to 100% as well as under a constant strain of 100% among different loading-unloading cycle (Supplementary Fig. 8, Supplementary Movie 1). Additionally, the IHPS unit can maintain its original shape without any sign of rupture under a strain of 100% in tension for 50 cycles, indicating excellent elastic recovery and resilience capability. During cyclic compression tests, each kind of hydrogel sample exhibited similar reproducible stress-strain curve patterns and can restore its original shape without fracture (Supplementary Fig. 9). The excellent mechanical robustness of these ionic hydrogels was mainly attributed to their abundant reversible ionic and hydrogen bonds within the hydrogel matrix networks[25], making them promising for potential applications in the fields of soft robotics and flexible electronics.

The flexible and stretchable nature of the consecutively-printed IHPS enables them to maintain stable electrical outputs under multiple deformation conditions, which was crucial for their potential wearable applications[26]. Figure 3d shows the $V_{oc}$ and relative resistance change ($\Delta R/R_0$) of the IHPS unit as the elongation ($\varepsilon_l$) gradually increased from 20% to 100%. The $V_{oc}$ remained basically unchanged even when the IHPS unit was stretched to an $\varepsilon_l$ of 60%, and only a slight decrease of 5.46% was observed upon further stretching to an $\varepsilon_l$ of 100%. The $\Delta R/R_0$ exhibits a step increase corresponding to the increase in strain (Fig. 3d), which increased to 94.04% at an elongation of 100%. The strain-induced resistance change of the IHPS unit was different compared with an ideal electroconductive resistor. This phenomenon might be caused by the inhomogeneous elongation of four types of hydrogels with different initial ion concentration and ion-selective efficiency in one unit[27,28], which still necessitates a comprehensive investigation in the future. Besides, during cyclic stretching testing, the consecutively-printed IHPS unit exhibited relatively stable voltage output, which showed a reduction of 2.52% after undergoing 1000 cycles of stretching-and-releasing ($\varepsilon_l = 100\%$) for 120 min (Fig. 3e).

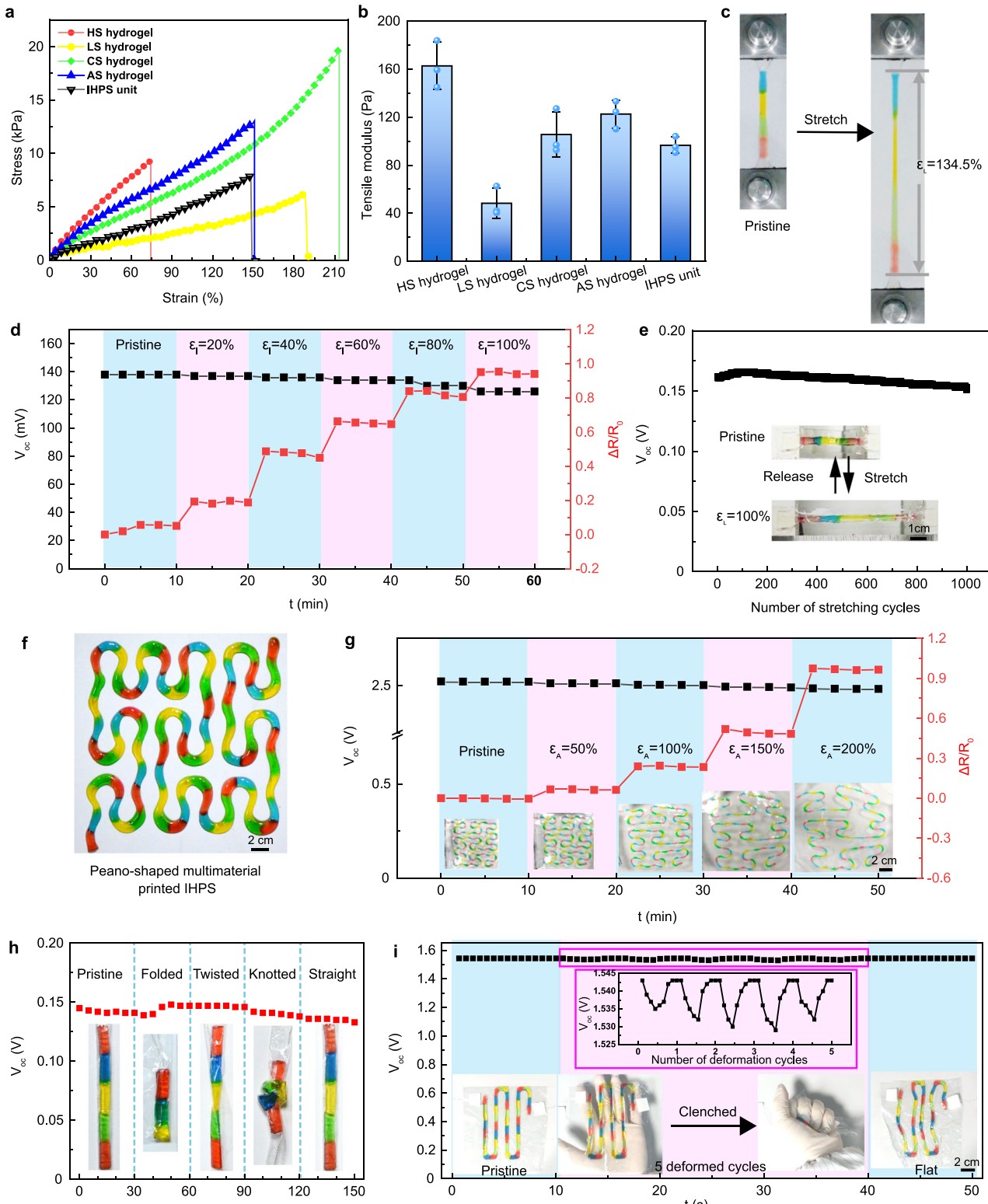

**Fig. 3 | The stretchability, flexibility, and robustness of consecutively-printed ionic hydrogel power sources (IHPS) under complex deformation conditions.** **a** The stress-strain curves and (**b**) tensile moduli (mean ± s.d., *n* = 3) of the consecutively-printed IHPS units and single-component printed hydrogel filaments. **c** Image of the deformed IHPS unit during the tensile testing. **d** $V_{oc}$ and the relative resistance value of the IHPS unit at $\varepsilon_I$ of 20%, 40%, 60%, 80%, and 100% (**e**) $V_{oc}$ profile of the cyclically stretched-and-released IHPS (1000 cycles with a maximum $\varepsilon_I$ of 100%). Insets show images of the IHPS unit at $\varepsilon_I$ of 0% and 100%. **f** The image, (**g**) $V_{oc}$, and the relative resistance value of a consecutively-printed Peano-shaped IHPS with 20 units in series at $\varepsilon_A$ of 50%, 100%, 150%, and 200%. **h** $V_{oc}$ profile of the consecutively-printed IHPS unit under folding, twisting, and knotting conditions. **i** $V_{oc}$ profile of the IHPS (10 units in series) during manual clenching-relaxation cycles. Source data are provided as a Source Data file.

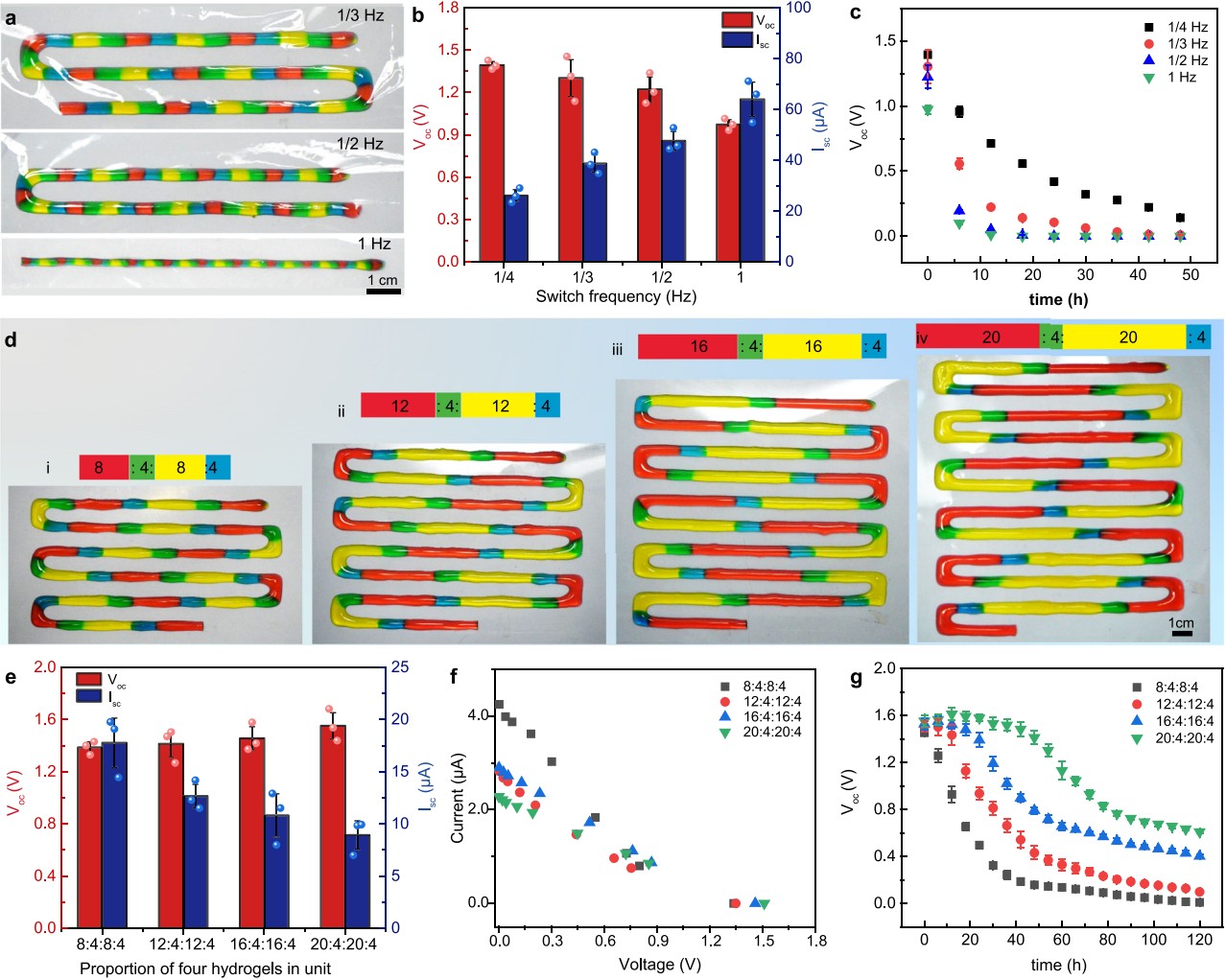

**Fig. 4 | Regulation of the switching frequency to modulate the proportion and obtain desired electrical characteristics of the consecutively-printed ionic hydrogel power sources (IHPS).** a Photographs, (**b**) $V_{oc}$, $I_{sc}$, and (**c**) power dissipation curves of the consecutively-printed IHPS with 10 units in series at a constant switching frequency of 1/3 Hz, 1/2 Hz, and 1 Hz (mean ± s.d., $n = 3$). d Photographs, (**e**) $V_{oc}$, $I_{sc}$, (**f**) current-voltage curves, and (**g**) power dissipation curves of the consecutively-printed IHPS containing 10 units in series with the HS:CS:LS:AS proportion varying from 8:4:8:4, 12:4:12:4, 16:4:16:4 to 20:4:20:4 (mean ± s.d., $n = 3$). Source data are provided as a Source Data file.

During each stretching cycle, the voltage periodically fluctuated within a relatively constant amplitude of 1–2 mV (Supplementary Movie S2).

Additionally, the IHPS can be easily printed onto a flexible substrate in a user-defined pattern, e.g., Peano-shaped curves (Fig. 3f). The $V_{oc}$ of the resultant IHPS with 20 units connected in series, remained stable during a maximum plane stretching strain ($\varepsilon_A$) of 200% as shown in Fig. 3g. The $\Delta R/R_0$ for the Peano-curved IHPS showed a nonlinear increase of *c.a.* 12%, 20%, and 55% corresponding to the $\varepsilon_A$ of 50%, 100%, and 150%, respectively. This phenomenon might be mainly attributed to the shape-induced deformation history of the Peano-curved power source during the plane stretching process, which can provide new insights to develop flexible power supply systems that are less susceptible to plane deformation.

Besides high stretchability, the consecutively-printed IHPS unit also exhibited high flexibility and robustness under various deformation conditions including manual folding, twisting, and knotting, as evidenced by the stable $V_{oc}$ with a fluctuation of less than 7% (Fig. 3h). The voltage drift during manual deformation mainly results from the unstable contact between the relatively rigid electrodes and soft hydrogel. Afterward, the IHPS unit can restore its initial shape and voltage output after deformation. To further confirm the robustness of the consecutively-printed IHPS, a 10-unit IHPS was manually distorted

and the electrical performance during deformation was recorded. Figure 3i shows the $V_{oc}$ profile of the IHPS, which underwent five manual clenching-relaxation cycles. During each cycle, the IHPS experienced a recoverable voltage fluctuation in a small range of 0.52%–0.91% (Supplementary Movie 3). These properties of the consecutively-printed IHPS in stable voltage output under various deformation conditions make them promising for potential applications in the fields of soft robotics and flexible electronics[29].

## Electrical modulation of the consecutively-printed IHPS

Since the proposed consecutive multimaterial printing strategy can easily and precisely control the proportion of each hydrogel component in a single power source unit by adjusting the switching frequency, it provides an innovative way to modulate the electrical performances of the resultant IHPS. Figure 4a presents the 10-unit IHPS consecutively printed at different switching frequencies of 1/3 Hz, 1/2 Hz, and 1 Hz (Supplementary Movie 4), respectively. As the switching frequency increased, the IHPS unit length was reduced from 36.12 ± 0.07 mm to 12.05 ± 0.05 mm and the corresponding filament width decreased from 4.02 ± 0.25 mm to 1.89 ± 0.16 mm (Supplementary Fig. 10). The $V_{oc}$ was reduced from 1.30 ± 0.13 V to 0.97 ± 0.03 V while the $I_{sc}$ significantly increased from

$38.74 \pm 3.40\,\mu A$ to $63.98 \pm 6.76\,\mu A$ (Fig. 4b and Supplementary Fig. 11). This was mainly due to the reduced internal resistance at a smaller IHPS length. In addition, the discharge rate of the IHPS was found to accelerate as the switching frequency increased from 1/4 Hz to 1 Hz as a result of the smaller ion amount and shortened ion transport pathways (Fig. 4c).

The electrical properties of IHPS were highly determined by the amount of ions inside the HS and LS hydrogel components, which offers a new opportunity to regulate power source discharge properties by changing the switching frequency. Figure 4d shows the consecutively-printed IHPS with 10 units in series when the switching frequency for CS and AS components was constantly fixed at 1/4 Hz while that for the HS and LS components was changed into 1/8 Hz, 1/12 Hz, 1/16 Hz, and 1/20 Hz (Supplementary Movie 5), respectively. It can be clearly seen that the CS and AS portions maintained the constant length of $12.04 \pm 0.15$ mm while the length of the HS and LS portions gradually increased from $24.16 \pm 0.21$ to $60.09 \pm 0.16$ mm. Consequently, the $V_{oc}$ remained relatively stable, while the $I_{sc}$ remarkably decreased from $17.79 \pm 2.36\,\mu A$ to $8.93 \pm 1.59\,\mu A$ due to prolonged unit length (Fig. 4e). The current-voltage curves of the consecutively-printed IHPS with varying HS and LS proportions exhibited similar and linear profiles with different slopes (Fig. 4f), confirming the scalability and tunability of their electrical properties. Moreover, the discharge rate of the IHPS was significantly decelerated by the increase in the length of HS and LS portions. As shown in Fig. 4g, the $V_{oc}$ of IHPS with an HS:CS:LS:AS portion ratio of 8:4:8:4 exhibited an exponential decline over time, whereas that of the IHPS with a portion ratio of 12:4:12:4 initially maintained a stable $V_{oc}$ (>90% of original $V_{oc}$) for $14.95 \pm 0.54$ h before experiencing an exponential decrease. Further increasing the HS and LS proportion to 16 and 20 resulted in a stable $V_{oc}$ of the consecutively-printed IHPS for more than $27.36 \pm 0.74$ h and $48.71 \pm 0.68$ h. Similarly, the time for the loss of 50% of the original $V_{oc}$ was also significantly prolonged from $14.35 \pm 0.26$ h for IHPS with a proportion of 8:4:8:4 to $84.24 \pm 0.63$ h (proportion of 20:4:20:4, Supplementary Movie 6). These phenomena were primarily attributed to the augmented ionic concentrations and elongated ionic migration pathways of both HS and LS portions. Notably, the $I_{sc}$ of the IHPS with a proportion of 20:4:4:4 was improved to $19.36 \pm 4.25\,\mu A$ by purely lengthening the HS portion in the unit, which was higher than that of the unit with a proportion of 20:4:20:4 due to the reduced internal resistance of the shortened LS hydrogel component. The half-life of the IHPS with a proportion of 20:4:4:4 was significantly extended by increasing the HS portion compared with that with a proportion of 4:4:4:4, which is consistent with our finding that the half-life can be extended with elongated ionic migration pathways (Supplementary Fig. 12). However, purely increasing HS portion in the unit did not show an obvious "plateau" region in the power dissipation profile as that of 20:4:20:4, which may be attributed to the lower LS proportion, resulting in a smaller volume and a shorter ion migration pathway. Additionally, it was found that elongating the ion transfer pathway by increasing the length of both CS and AS portions can effectively decrease the discharge rate while maintaining initial $V_{oc}$ magnitude (Supplementary Fig. 13), demonstrating a high degree of tunability in their power dissipation profiles.

### Multimaterial printing and automatic collection of the IHPS

The consecutive multimaterial printing strategy can be further integrated with automatic substrate feeding and collecting functions to realize assembly-free fabrication of IHPS with a large number of units. As shown in Fig. 5a, the serpentine-configured IHPS units were consecutively printed onto the flexible and moving membrane in series, cross-linked by a UV spot light, and finally collected on a rotating roll (Supplementary Movie 7). It is noteworthy that the ion transportation within the consecutively-printed IHPS instantly occurs once crosslinked, resulting in total voltage

dissipation throughout the collection process of the IHPS roll and thus leading to inevitable energy loss. To mitigate power loss, we devised a thermostatic chamber with a low temperature of 4 °C to slow down the movement of ions (Supplementary Fig. 14). As shown in Fig. 5c, the $V_{oc}$ of the IHPS roll containing 1000 units in series was *c.a.* 141.01 V when collected at 4 °C, which is close to the predicted voltage obtained from the number of units in relation to the series voltage in Fig. 2i for a 1000-unit IHPS (*c.a.* 142 V). In contrast, the IHPS roll with 1000 units connected in series exhibited a $V_{oc}$ of *c.a.* 116.5 V when collected at 25 °C. The findings demonstrated that the low-temperature chamber can significantly reduce power dissipation during the printing process. As a simple demonstration, the consecutively-printed IHPS containing 1000 units in series enabled to power a light-emitting "XJTU" pattern with 21 diodes (Fig. 5d).

To improve the printing efficiency for IHPS with large unit numbers, the printheads can be specifically designed to have multiple arrayed outlets, and the UV spot light was replaced by a surface UV light, which enabled the simultaneous printing and crosslinking of parallel IHPS as shown in Fig. 5e. For this configuration, the printhead was kept still while the flexible membrane was constantly moved by the rotation of the driving roller[30,31]. The resultant parallel power sources were serially connected via conductive circuits for large voltage output (Fig. 5f and Supplementary Fig. 15). Figure 5g presents the arrayed multimaterial printheads with 2, 4, 8, and 16 outlets. For each hydrogel ink, it will flow through the same channel distance to reach the corresponding outlets, which facilitated identical flow distribution and stable consecutive printing of IHPS as shown in Fig. 5h. The resultant IHPS printed by the printhead with 16 outlet arrays showed a similar electrical performance to those derived from the single-outlet printhead in terms of $V_{oc}$, $I_{sc}$, and power dissipation rate (Fig. 5i, j). These results suggested that the inter-circuit connection among parallel IHPS has little effect on the electrical performance. More importantly, the time cost of printing an IHPS with a large number of units can be exponentially reduced. As an example demonstration, the IHPS with 1600 units can be rapidly printed within 0.44 h by using a printhead with 16-arrayed outlets, while that for the single-outlet printhead was approximately 7.11 h. The developed consecutive multimaterial printing strategy provide an efficient way to realize the fabrication of hydrogel-based salinity-gradient power sources with the highest voltage output of 208.01 V in less than half an hour (Fig. 5k, Supplementary Movie 8-9), which is 1.89 times higher than the maximum voltage value of 110 V for the electric eel-inspired IHPS[12].

The $I_{sc}$ of the consecutively-printed IHPS can also be improved by connecting separate IHPS units in parallel with inter-circuits (Fig. 6a). As shown in Fig. 6b, the $I_{sc}$ of the IHPS increased linearly with the increase of unit number in parallel, which reached $66.75 \pm 10.08\,\mu A$, $123.50 \pm 18.95\,\mu A$, $224.01 \pm 41.68\,\mu A$, and $426.75 \pm 74.47\,\mu A$ in corresponding to the parallelly-connected unit number of 2, 4, 8, and 16 arrays respectively. In such a case, the $V_{oc}$ of the resultant IHPS remained unchanged, similar to that for a single IHPS unit.

### Proof-of-concept applications of the IHPS

Taken together, the presented consecutive multimaterial printing strategy offers a highly efficient and powerful approach to fabricate user-specific IHPS with tunable electrical properties through multiple combination strategies, such as varying the switching frequency or inter-circuit connection configuration. For example, an IHPS with a $V_{oc}$ of 3.55 V and an $I_{sc}$ of 1.02 mA can be flexibly printed in parallel by using a 16-array printhead (Fig. 6c). For each parallel power source line, a total of 30 IHPS units were printed in series at a switching frequency of 1/2 Hz. The resultant power source was successfully used to activate a poly(3,4ethylenedioxythiophene)/poly(styrenesulfonate) (PEDOT: PSS)-based electrochromic device. As shown in Fig. 6d, the flower pattern underwent a rapid transition to bright blue within 10 s when the electrochromic device was connected to the IHPS. Conversely, the

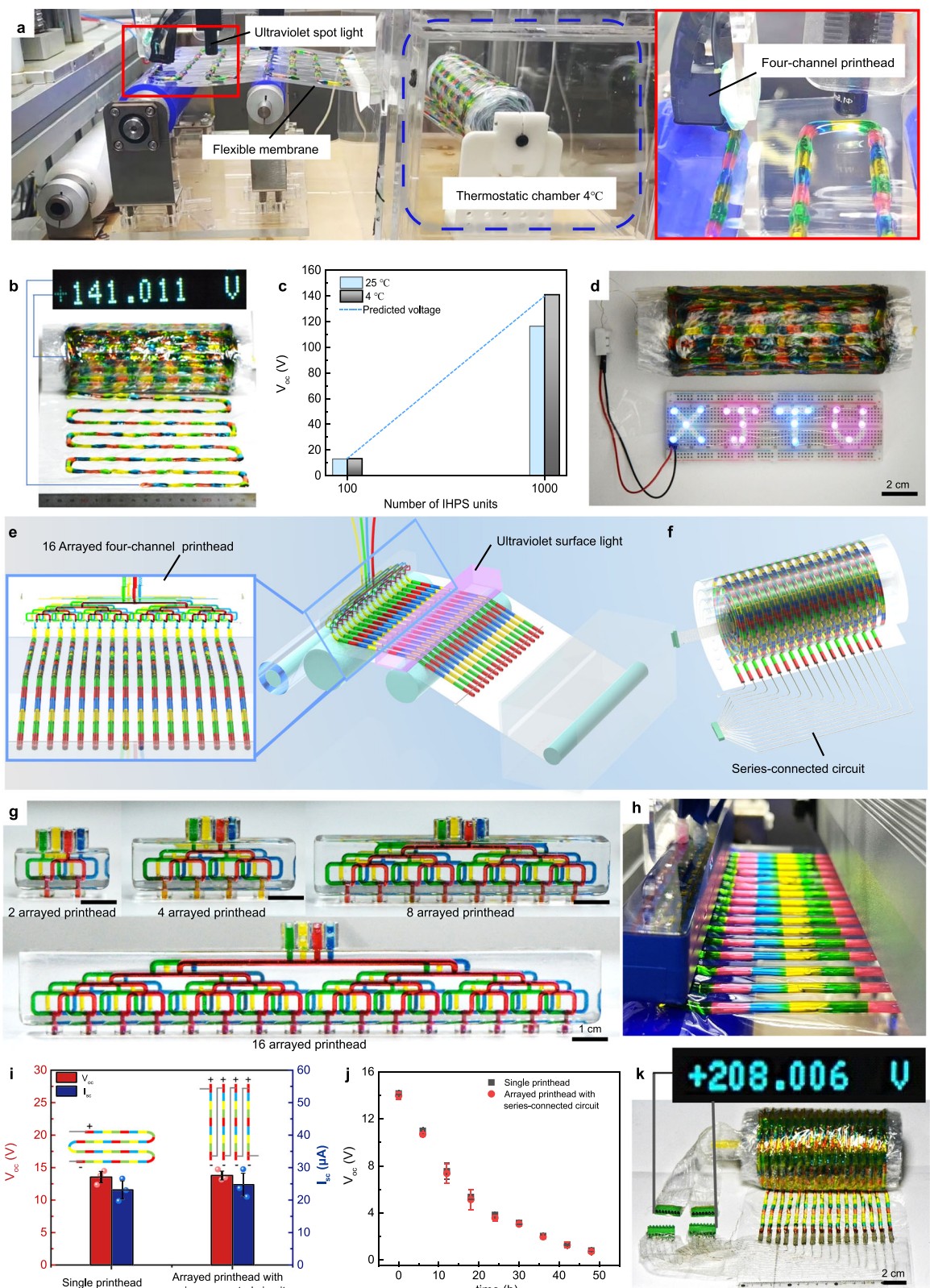

blue flower pattern can recover to its original light gray color as the electrochromic device was connected in reverse to the IHPS (Supplementary Movie 10).

As another typical demonstration, IHPS were consecutively printed at a switching frequency of 1/2 Hz for 15 units in series, and a total of 16 arrays were simultaneously printed and connected in parallel, which generated a $V_{oc}$ of 1.53 V and an $I_{sc}$ of 1.10 mA. Due to the unique

flexibility and stretchability, these IHPS filaments can be manually interlaced into a four-strand braid configuration (Fig. 6e, Supplementary Fig. 16), which had little effect on the $V_{oc}$ and $I_{sc}$ of the weaved IHPS (Fig. 6f). The resultant IHPS was worn on the wrist and successfully powered a digital wristwatch as shown in Fig. 6g, demonstrating good flexibility as a soft power source for potential wearable applications in flexible electronics.

**Fig. 5 | Fabrication of high-voltage ionic hydrogel power sources (IHPS) with many units in series by automated consecutive multimaterial printing and collecting strategy. a** The automatic consecutive multimaterial printing process was complemented with a roll-to-roll collection module. **b** The photograph of the resultant IHPS roll with 1000 units in series, which was consecutively printed at a switching frequency of 1/4 Hz. **c** $V_{oc}$ of the consecutively-printed IHPS collected at a temperature of 25 °C and 4 °C, respectively. **d** The illumination of 21 LED bulbs (1.3 mW) arranged in the "XJTU" pattern was accomplished by utilizing the consecutively printed IHPS roll with 1000 units in series. **e** The schematic of consecutive multimaterial printing process for parallelly-configured IHPS utilizing an arrayed multimaterial printhead. **f** Inter-circuits were designed for IHPS to achieve a series-connected configuration with enhanced voltage output. **g** The photograph of arrayed multi-channel printheads with 2, 4, 8, and 16 outlets. **h** The snapshot of consecutive multimaterial printing process using a 16-arrayed multimaterial printhead. **i** $V_{oc}$, $I_{sc}$, and (**j**) power dissipation characteristics (mean ± s.d., $n = 3$) of the printed IHPS consisting of 96 units connected with inter-circuits using an arrayed multimaterial printhead were found to be comparable to those obtained from a single printhead with the same unit number in series connection configuration. **k** $V_{oc}$ measurement results showed that the printed IHPS containing up to 1600 units in series achieved an impressive voltage output reaching up to 208.01 V. Source data are provided as a Source Data file.

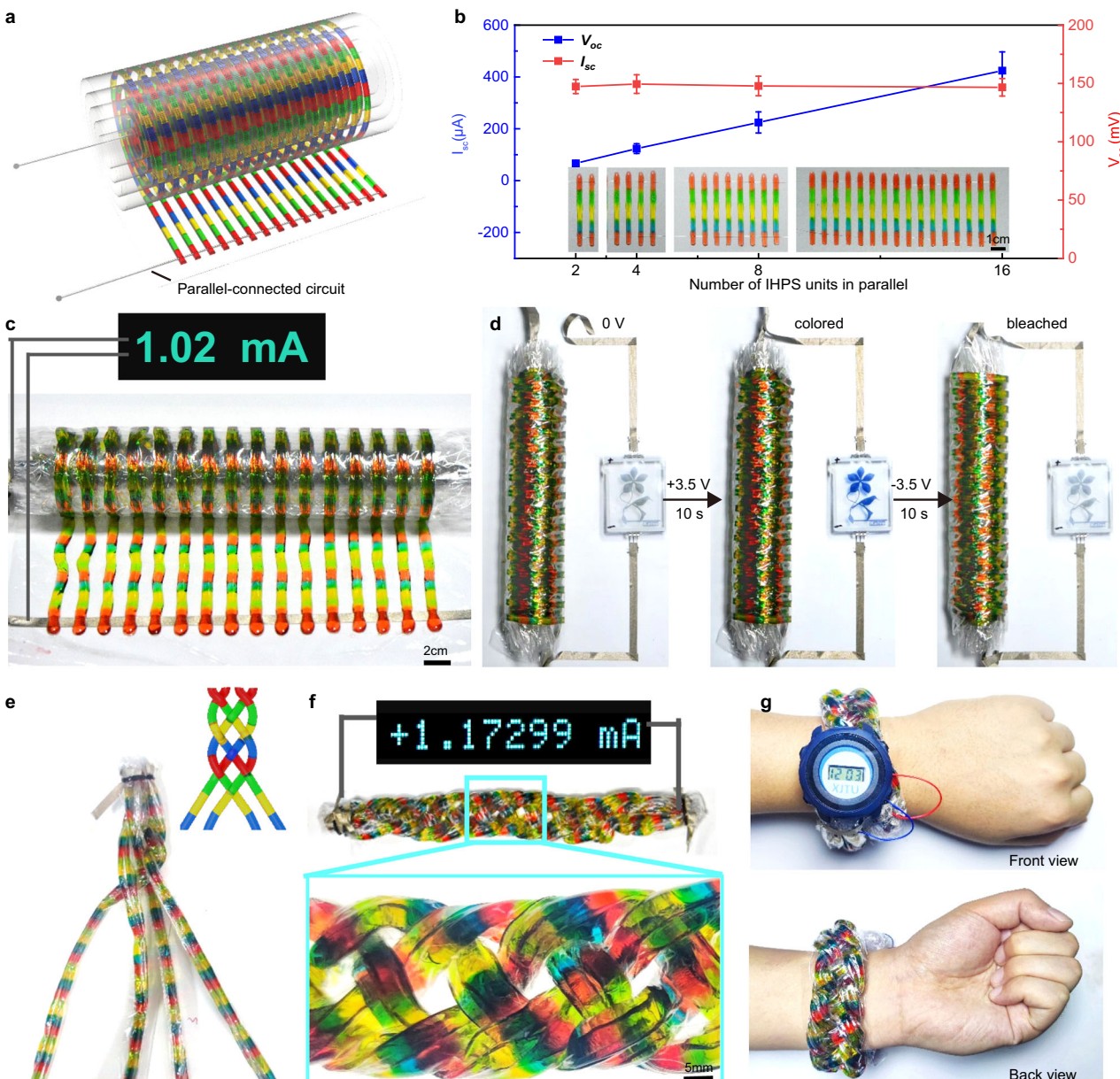

**Fig. 6 | Improving the $I_{sc}$ of the consecutively-printed ionic hydrogel power sources (IHPS) by connecting arrayed IHPS filaments in parallel. a** The schematic illustration to connect arrayed IHPS filaments in parallel via inter-circuits. **b** Effect of the number of arrays in parallel on the $V_{oc}$ and $I_{sc}$ of the IHPS (mean ± s.d., $n = 3$). **c** The photograph and $I_{sc}$ of a 16-paralleled-connected IHPS with 30 units in series. **d** Activation of PEDOT: PSS-based electrochromic devices by the as-printed IHPS. **e** The photograph depicts the weaving process of a consecutively-printed IHPS with 15 units in series and 16 arrays in parallel, while the inset provides a schematic representation of its braided structure. **f** The photograph and $I_{sc}$ of the woven IHPS. **g** The woven IHPS can be used as a wristband to power a digital wristwatch. Source data are provided as a Source Data file.

## Discussion

To further underscore the merits of our research, we conducted a comprehensive comparison between the performance of the presented IHPS and the existing ones, including electrical efficiency, stretchability, working temperature, and fabrication versatility as shown in Table 1. Firstly, the existing strategies for power generators harnessing ion gradients commonly relied on the manual assembly of two or multiple layers of different hydrogel components, which highly affected the interface contact stability of the resultant IHPS, especially when working under considerably deformable conditions. In this study, a consecutive multimaterial printing strategy was developed to continuously fabricate the IHPS with seamless interface. This feature endowed the resultant power source excellent stretchability and flexibility. The consecutive IHPS filaments exhibit a maximum stretchability of up to 137% and remain stable voltage output during over 1000 stretching cycles at an elongation of 100%. The presented power source demonstrates superior resilience to a wider range of large deformation patterns, including bending, folding, knotting, and clenching. In contrast, other hydrogel-based power sources exhibit limited functionality under only folding[13] or bending[14], primarily due to the unstable interface between adjacent hydrogel components during extensive deformations.

Secondly, the presented multimaterial printing strategy is highly efficient and facilitates the future scalability of the IHPS due to assembly-free characteristics. A maximum voltage output of 208.01 V can be easily achieved by using a 16-arrayed multimaterial printhead to print the IHPS with 1600 units in 0.44 h, which provides highly efficient fabrication strategy for hydrogel-based power sources harnessing ion-concentration gradients with flexibility and stretchability. Additionally, the IHPS filaments can be interconnected in parallel to improve electrical current output.

Thirdly, the proposed automatic consecutively-printing strategy facilitates the efficient and facile fabrication of high-voltage IHPS with user-specific electrical properties, making it a valuable contribution to the field. Additionally, the consecutive printing technique enabled to dynamically and flexibly regulate the ratio of four components during the fabrication process of the IHPS by modulating the switching frequency. The electrical properties of IHPS were highly determined by the amount of ions inside the HS and LS hydrogel components, which offers a feasible method to regulate power source discharge properties by changing the switching frequency. By modulating the IHPS unit proportion, the half-life of the resultant IHPS was prolonged to over 84 h. However, it poses great challenge for the existing manually-assembled fabrication techniques of hydrogel-based power sources to flexible regulate the half-life of the power source system in terms of efficiency and complexity.

Finally, the consecutively-printed IHPS can also work under a wide temperature range (−20 °C–90 °C). Typically under the conditions of −20 °C, the HS and LS hydrogels exhibited exceptional anti-freezing properties due to the presence of lithium chloride[32,33] and/or glycerol[34], whereas the CS and AS hydrogels experienced partial freezing with weakly bound water and unfrozen water inside the hydrogel matrix (Supplementary Fig. 17)[35]. The presence of unfrozen water along with the seamless interface of the as-printed ionic hydrogel filament facilitates the stable connection and the continuous ionic transportation for stable voltage output even at a low temperature of −20 °C. Nevertheless, the aforementioned IHPS with multilayer structure might fail to discharge below freezing temperatures since the contact interface of the multilayer hydrogel was destroyed by water precipitation from the hydrogel matrix which prevented the formation of an effective ion transport.

Although promising, the IHPS inspired by electric eels are still in their nascent stage and the application scenario in the real world is very limited. In a recent study, Zhang et al.[36] have developed a biocompatible microscale IHPS which served as ionic current sources to

**Table 1 | Comparison of parameters from ionic power sources inspired by electric eels**

| Ionic power source | Power source structure | Stretchability | Maximum $V_{oc}$/ V | Peak $I_{sc}$/ mA | Half-life/ h | Operation temperature/ °C |
|---|---|---|---|---|---|---|
| Surface-printed power source[12] | Rigidly assembled double-layer structure | N/A | 110.0 | 0.01 | ~0.5 | Ambient temperature |
| Paper-gel power source[14] | Rigidly-assembled multilayer structure | N/A | 3.2 | 8.5 | ~0.08 | Ambient temperature |
| Microfluidics-based flexible power source[13] | Flexibly-assembled double-layer structure | N/A | 73.3 | 0.1 | ~0.86 | 0 –90 |
| Microscale ionic power source[36] | Two-step processed continuous structure | N/A | 2.0 | $2.2 \times 10^{-3}$ | ~0.5 | 25 –37 |
| This work | Consecutive structure | ~137% | 208.0 | 1.1 | ~84.24 | −20 –90 |

Half-life means the time for the voltage to decay by half.

modulate the activity of neuronal network in neural microtissues in vitro. In terms of the safety and eco-friendliness of the soft hydrogel-based power sources, IHPS have shown their potential as biocompatible power sources for biological stimulation in the next-generation bio-hybrid interfaces, implants, and synthetic tissues. However, the main challenge for IHPS is their relatively poor power output in comparison to conventional energy storage systems. The low output power of IHPS was mainly caused by the lower through-put of ion transport flux, which was attributed to the limited selectivity and the high internal resistance of the hydrogel matrix during ion transportation[14,37]. It has been proven a feasible way to improve the power output of the IHPS by reducing their internal resistance by incorporating conductive materials such as PEDOT:PSS[13] and MXene[38] into the hydrogel matrix as well as shortening the ion conductive pathways by further increasing the switching frequency. Apart from this, there is a need to develop novel ion-selective hydrogels that possess high permeability, exceptional ion selectivity, and low internal resistance[39]. For example, constructing aligned microchannels inside the ion-selective hydrogels in parallel to the ion transportation direction might enhance the ion diffusion rate by decreasing ion diffusion pathway[40,41]. One of the major challenges for the present IHPS is the gradually decreased voltage output, as the ion gradients decrease between the HS and LS hydrogels during discharging. Future efforts to address this challenge could be the integration of efficient solar desalination processes[42–44] that can regenerate ion-concentration gradients with the RED-based power sources, thereby forming closed-loop-systems capable of establishing sustainable salinity power generators for long-term use. In the future, it might be possible to construct bioactive IHPS in vitro by fabricated gene edited electrocytes with serial-connected arrangement mimicking the structure in EOs via cell printing technique. It is believed that with the rapid development of ion-selective materials and continuous breakthroughs in advanced fabrication strategies, the electric-eel-inspired IHPS would mature into a viable soft power source and accelerate the innovative development of flexible electronics, soft robotics, and implants.

Here we proposed a consecutive multimaterial printing strategy to automatically fabricate IHPS with high flexibility and stretchability by mimicking the power generation mechanism of biological organs inside electric eels. Four kinds of IHPS inks were optimized to have similar rheological properties to facilitate consecutive printing of IHPS filaments with seamless bonding interface, uniform size, and morphology, which showed high stretchability up to 137% and stable voltage output capability under large deformation and cyclic stretching conditions. Interestingly, the presented printing strategy enabled to flexible modulate the electrical property of the IHPS unit by regulating the switching frequency and subsequent the ratio of hydrogel components. Automatic printing and collecting platforms were successfully developed to realize assembly-free fabrication of IHPS with little voltage loss in one configuration. We further demonstrated that the presented strategy can be easily scaled up by using a multi-channel printhead array for parallel printing of IHPS, enabling to achieve a high voltage of 208 V within less than 30 min. The developed multimaterial printing technique provided a highly efficient fabrication strategy to realize such high voltage output by hydrogel-based power sources harnessing ion-concentration gradients. The as-printed IHPS filaments can be flexibly connected in series or in parallel to activate electrical devices or further weaved into a wearable wristband to power a digital wristwatch, demonstrating great promise for potential applications in various soft and flexible power source scenarios.

## Methods

### Materials
Lithium chloride, acrylamide, (3-acrylamidopropyl)trimethyl ammonium chloride, 2-acrylamido-2-methylpropane sulfonic acid, HEC

(average Mv ~1,300,000) were purchased from Sigma-Aldrich (China). Acrylamide/N, N′-methylenebisacrylamide with a mole ratio of 37.5:1 (henthforth "bis") and glycerol were bought from Energy Chemical (China). 2-Hydroxy-2-methylpropiophenone (henthforth "photo-initiator"), hydroquinone, and PEO (average Mv~4,000,000) were bought from Aladdin (China).

All the ionic hydrogel precursors were prepared by dissolving corresponding reagents into ultra-pure water. HS precursor: 6.0 mol L$^{-1}$ lithium chloride, 0.054 mol L$^{-1}$ bis, 4.38 mol L$^{-1}$ acrylamide, 0.007 mol L$^{-1}$ photoinitiator. LS precursor: 0.015 mol L$^{-1}$ lithium chloride, 0.062 mol L$^{-1}$ bis, 4.10 mol L$^{-1}$ acrylamide, 3.4 mol L$^{-1}$ glycerol, 0.015 mol L$^{-1}$ photoinitiator. AS precursor: 2.0 mol L$^{-1}$ (3-acrylamido-propyl)trimethyl ammonium chloride, 0.034 mol L$^{-1}$ bis, 2.75 mol L$^{-1}$ acrylamide, 0.012 mol L$^{-1}$ photoinitiator. CS precursor: 2.0 mol L$^{-1}$ 2-acrylamido-2-methylpropane sulfonic acid, 0.055 mol L$^{-1}$ bis, 1.90 mol L$^{-1}$ acrylamide, 0.012 mol L$^{-1}$ photoinitiator. To enhance stability of CS ink, hydroquinone was introduced into the CS precursor as a polymerization inhibitor at a concentration of 0.15 mg/mL[45]. To enable consecutive printing, the resulting inks were formulated by incorporating HEC rheology modifiers at concentrations of 1.30%, 1.52%, 1.35%, and 1.82% into the HS, LS, CS, and AS precursors, respectively, while maintaining a constant PEO concentration of 0.05%. To distinguish the HS, LS, CS, and AS hydrogel inks, water-soluble red, yellow, green, and blue dyes were added to the resultant inks, respectively.

### Rheology regulation of IHPS inks
To achieve efficient switching of four IHPS inks, the concentrations of HEC and PEO added to the precursor solution were optimized to regulate the four inks to exhibit similar rheological properties and printability. Rheological properties of the IHPS inks with different concentrations of HEC and PEO were measured using a rheometer (Anton Paar MCR 702, Ashland, US) with parallel plates of 25.0 mm in diameter and a plate-to-plate gap distance of 1.0 mm. The shear stress was measured with the shear rate varying from 0.01 s$^{-1}$ to 200 s$^{-1}$ at a temperature of 25 °C, whereas oscillatory measurements were carried out at a frequency of 1 Hz within the angular frequency range of 0.1–100 rad s$^{-1}$ [46]. All rheological measurements were performed immediately after homogenization.

### Consecutive multimaterial printing of IHPS
Figure 1e shows the house-made consecutive multimaterial printing system that mainly consists of five components: a pressure controller module (FLOW-EZ, 0–700 kPa, Fluigent, France) for ink extrusion, a multichannel solenoid valve sets (4V-220/08, Zhengtai, China) for ink switching, a specially-designed multimaterial printhead (Yichangtai, China) for ink consecutive printing, a UV spot light (H-CM-DGY-04, Shenzhen Heshengbang Co., Ltd., China) for ink cross-linking and an XYZ motion platform for the formation of designed structural patterns.

To realize controllable consecutively printing of IHPS, the widths of the average and switching points in IHPS filaments were measured at pressures ranging from 80 kPa to 180 kPa and stage moving speeds from 2.0 mm s$^{-1}$ to 5.0 mm s$^{-1}$. To measure the average width of the printed IHPS filament, sampling locations were chosen at 1/4, 1/2, and 3/4 lengths of HS, CS, LS, and AS hydrogel portions, respectively. The width at the switching point was measured at the interface between distinct hydrogel portions. The uniformity of the printed IHPS filament was calculated as follows[20,47]:

*The uniformity of printed ionic power source filament* $= d_{switch}/\bar{d}$

$$(1)$$

Where, $d_{switch}$ represents the width of IHPS filaments at the material switching points, $\bar{d}$ denotes the average width of the printed IHPS filaments. The measurements were taken from distinct samples.

## Electrical characterization of the IHPS

To accurately measure the electrical performances of the consecutively-printed IHPS, two silver electrodes were embedded into the HS hydrogels before cross-linking. The $V_{oc}$, $I_{sc}$, internal resistance, and dissipation of the consecutive IHPS were tested with a Keithley electrometer (6517B). The normalized current-voltage curves were obtained by connecting the consecutively-printed IHPS with resistors ranging from 1 kΩ to 1 MΩ. And the normalized voltage-current values of a single IHPS unit were then calculated by normalizing the number of units connected in a series or parallel in the power sources. The measurements were taken from distinct samples.

To investigate the effect of the assembly-free consecutive printing strategy on the impedance of the IHPS unit, four types of hydrogel were multimaterial printed in a mold with a width of 10 mm to ensure that each of the hydrogel dimensions of about 10 mm in length, 10 mm in width, and 2 mm in thickness. In comparison, the control group with four separate hydrogel portions assembled in two layers was produced by the microfluidic-based perfusion strategy each hydrogel with a dimension of about 10 mm in length, 10 mm in width, and 2 mm in thickness[13]. The electrochemical property of the as-fabricated IHPS unit was characterized by using an electrochemical workstation (MetrohmAutolab PGSTAT302N, Switzerland) with a conventional two-electrode method in the swept range from 0.01 Hz to 3 MHz.

## Water retention performance of the IHPS

The water retention performance of four types of hydrogel materials and the consecutively-printed IHPS unit with a fixed length of 48.0 mm was investigated under different temperature and humidity conditions (25 °C and 45% RH, 60 °C and 45% RH, 25 °C and 90% RH) for 12 h. The initial water content of the samples was set as 100%. The water content of the hydrogels at different time point was further calculated by the following equation:

$$The\ water\ content = m_t - m_{dry}/m_0 - m_{dry} \times 100\% \qquad (2)$$

Where, $m_0$ is the initial weight of IHPS units, $m_{dry}$ is the dry weight of IHPS units after being stored at 90 °C for 12 h, and $m_t$ is the weight of IHPS units incubated at different environment for 12 h.

## Mechanical testing of the consecutively-printed IHPS

The mechanical properties of the consecutively-printed IHPS unit and four types of hydrogel filaments were tested using an electro-mechanical universal testing machine (103 A, Wance, China). The four hydrogel samples and the IHPS unit were initially printed with a length of 48 mm, secured on both ends of the clamp, and subjected to stretching at a crosshead speed of 1 mm s⁻¹. A digital camera (Nikon, D7100, Japan) was used to keep track of the entire stretching process for the printed IHPS filaments. The stress-strain curves were calculated from load-displacement measurements, and the tensile modulus was determined by the slope of the linear region of the stress-strain curves. For the cyclic tensile loading-unloading tests, the four types of hydrogel filaments with an initial length of 48 mm were stretched to a maximum strain of 50% for 50 cycles by using a 10 N load cell at a crosshead speed of 4 mm s⁻¹. The tensile properties of the consecutively-printed IHPS units were further investigated by gradually increasing the tensile strains from 20% to 100% and periodically subjected to a strain of 100% for 50 cycles at a crosshead speed of 4 mm s⁻¹. The cyclic compression tests were conducted on the four types of cylindrical hydrogel samples with a diameter of 13.0 mm and a height of 6.5 mm. The samples were subject to a strain of 50% for 50 cycles by using a 50 N load cell at a crosshead speed of 1 mm s⁻¹.

## Electrical stability of the consecutively-printed IHPS

The $V_{oc}$ and relative resistance changes of the IHPS unit filament, which was consecutively printed and packed with a stretchable elastomer (4905, 3 M), were measured as the indicator of electrical stability using the Keithley Digital Multimeter under a gradually stretching condition. The relative change in resistance was determined using a previously established method[24]. The durability was further tested by cyclic stretching-and-releasing test for 1000 cycles at the maximum $\varepsilon_l$ of 100% and a speed of 10 mm s⁻¹, and the corresponding $V_{oc}$ was also measured using a Keithley Digital Multimeter.

For the planar stretching condition, a 20-unit IHPS filament was consecutively printed onto the stretchable elastomer with a Peano-shaped configuration. The printed Peano-shaped IHPS was fixed with clamps and subjected to gradually increasing $\varepsilon_A$ to 50%, 100%, 150%, and 200% for the measurement of $V_{oc}$ and relative resistance changes.

For the folding, twisting, and knotting conditions, the consecutively-printed IHPS unit was manually folded, twisted, and knotted for 30 s, respectively, and the $V_{oc}$ during the deformation process was recorded. Furthermore, the serpentine-configured IHPS with 10 units in series was clenched manually for five cycles, and the $V_{oc}$ was recorded.

## Multi-channel printhead array for parallel printing of IHPS

To fabricate IHPS more efficiently, the four-channel printhead was extended to the multi-channel arrayed printhead. The arrayed multi-material printheads were designed with hierarchical two-branched structures to ensure the equal flow rate of the inks at each branched channel and outlet. The arrayed multimaterial printhead was 3D-printed with a transparent resin by Yichangtai Intelligent Machine Co., Ltd. (China). For optimal printing performance, the air pressure settings for multimaterial printheads with 2-, 4-, 8-, and 16-arrays were adjusted to 150, 180, 240, and 320 kPa respectively to overcome increasing flow resistance with increased channel path.

To achieve high-voltage or high-current efficiently, inter-circuits were specifically designed to connect the arrayed IHPS in parallel or in series. The series-connected circuit contains two electrode systems, in which silver wires were arranged at an interval of 10 mm to connect the adjacent power source filament positively and negatively via multi-channel terminals. The series-connected circuit was embedded in the beginning and end of the IHPS which connected the arrayed-printed IHPS in series for improved voltage. For the parallel-connected circuit, the electrodes were embedded in the first and last IHPS unit of the arrayed IHPS for improved current.

## Data availability

All relevant data supporting the findings of this study are available within the article and its Supplementary Information files as well as Source Data. Source data are provided with this paper.

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

## Acknowledgements

J.H. thanks the following programs for the financial support: National Natural Science Foundation of China (52125501), the Key Research Project of Shaanxi Province (2021LLRH-08), the Program for Innovation Team of Shaanxi Province (2023-CX-TD-17), the Research and Development Project of Shaanxi Province (2023-KXJ-174) and the Fundamental Research Funds for the Central Universities.

## Author contributions

J.H. and P.H. conceived and designed the work. P.H. and Z.Q. designed the multimaterial printer. P.H. designed experiments, conducted the experiments, and completed related data analysis and interpretation. J.Y. contributed to the multimaterial printing and

collection of the IHPS, data process, and visualization. J.H. and D.L. supervised the study. P.H. wrote the manuscript. J.H., Z.M., and P.H. edited the manuscript.

## Competing interests

The authors declare no competing interests.
