## [Peer Review File · Nature Communications]

Consecutive multimaterial printing of biomimetic ionic hydrogel power sources with high flexibility and stretchabilityREVIEWER COMMENTS

Reviewer #1 (Remarks to the Author):

The authors reported a novel consecutive multimaterial printing strategy to fabricate flexible battery filaments for applications. The design can assemble the ionic hydrogel batteries in 30 mins with a high voltage of 208 V. In addition, this strategy can be applied on various flexible power device applications. The result is good, and the discussion is very systematic. Therefore, this paper can be accepted on Nature communications after addressing the following questions.

1. Detailed discussions for the anti-freezing mechanism of the hydrogel electrolyte should be provided, which can not be just attribute to the seamless interface.
2. The water retention performance of the ionic hydrogel should be characterized at different environment such as humidity and temperature.
3. For mechanical strength characterizations, the cycling compressive stress-strain curves and cyclic tensile loading-unloading test for the hydrogel electrolyte can be provided.
4. FTIR, Raman or other spectra characterizations can be provided to illustrate the interactions in the hydrogel for better performance design.
5. Can the length of each section in the hydrogel battery influence the performance?

Reviewer #2 (Remarks to the Author):

I thoroughly enjoyed reading this well-arranged paper. Hydrogels are become very hot topic and show their broad applications in various fields, including batteries and sensors. The manuscript underscores the distinct advantage of direct printing of multi-materials over the conventional method of transferring prefabricated hydrogels into molds. To realize the consecutive printing process, four types of ionic hydrogel inks were developed, resulting in seamlessly bonded battery filaments. The filaments exhibit stable voltage outputs during repeatable stretching, culminating in an impressive voltage of 208 V. As this work details the printing of hydrogel and its impact on salinity power, I believe it is fit to Nat comm., pending minor revision.

The following issues need to be addressed:

- (1) The demonstrated batteries appear to be one-time use. Is there potential for enhancing durability by implementing a sustainable salt gradient, akin to the strategy proposed in *Energy Environ. Sci.* 2017, 10, 1923.
- (2) I noticed that the current-voltage curves were obtained by connecting hydrogel batteries using resistors. Could you furnish a typical IV curve by scanning voltage with an electrochemical workstation?
- (3) The claim of printed batteries achieving a maximum voltage of 208 V and a half-life of 84 h requires evidences to bolster credibility.
- (4) It remains unclear whether the printed hydrogel batteries are sealed. Will dehydration happen during measurement?

Reviewer #3 (Remarks to the Author):

In this work, the authors present a new fabrication approach and overall morphology for a hydrogel-based ion-gradient electrical power generation/storage ("battery") scheme first reported by Schroeder et al in 2017 (ref. 12). They have engineered the system such that they can print continuous multi-component fibers that each consist of the repeating series arrangement of four distinct hydrogels necessary to produce the intended power transduction effect. The resulting fibers are reasonably durable and flexible; a single repeating unit can be stretched substantially (to a strain of 150%, corresponding to a stretch ratio of 2.5 times the original length) and twisted and knotted in various arrangements without breaking. The authors also explored printing in a Peano curve morphology, which allows substantially more deformation. They explored in detail the effects of modulating the length of the various hydrogel units (framed in the paper as changing the

switching frequency of their printing setup) on the voltage and current generation capacity and discharge kinetics of the system, producing some interesting results. They finally demonstrated that they can exploit the flexibility, fiber geometry, and electrical modularity of this scheme to power a wearable device (an LCD watch).

The authors have done a very thorough job of engineering this system for production, which involves complex multimaterial printing and curing on a flexible substrate that is continuously moved in a roll-to-roll system; this sort of demonstration is of definite value to this field, as it represents a step toward practical implementation at scale. The data in the work supports the claims of the paper; I have not found flaws in the analysis or data interpretation. The methods are generally presented clearly and in detail (though I have a minor comment about this below). I accordingly recommend this work for publication after a few very minor revisions are made for clarity's sake, and I congratulate the authors on what they've been able to accomplish here.

I will now detail the minor revisions, and also suggest some possible avenues of further inquiry that the authors could pursue in future work, since they seem well-set-up to do so.

The minor revisions:

1. The authors must include the full compositions of the prepolymer solutions in their Methods section, rather than saying that they were prepared "according to the previously established protocols" and referring to reference #13. This is for two reasons: first, it's a basic courtesy to save the reader from having to look up another paper, but secondly because the paper referenced is not open-access like this one is; readers may have access only to this manuscript.

2. I am assuming (and I think I can see visually) that the printed fibers in Figures 3h-i and 6e-g are electrically isolated from one another using clear plastic wrap in order to prevent different parts of the hydrogel assembly from short-circuiting with each other via the flow of ions. This could be easily missed and should be both explicitly mentioned and explained in the text of the paper.

3. I have an issue with how the authors have worded lines 475-477 of the discussion section of the paper: "Additionally, incorporating conductive nanomaterials such as MXene, carbon nanotubes into the hydrogel matrix with designed microstructures such as aligned microchannels and nonporous structures can also enhance the ion flux with reduced internal resistance." First, I believe the authors meant to write "nanoporous" instead of "nonporous". More generally, all the referenced papers (refs 28-35) describe innovations in membrane materials in order to achieve high selectivity, high overall permeability, and low resistance, as the authors say, and they could in fact be useful in a scheme like the authors have described. However, I think the authors need to make clear that the inclusion of the types of nanomaterials and membrane structures described in these papers would *replace* the function of the cation and anion selective gels, rather than enhancing their functionality as dopants in the gel. (A lot of the improvement in resistance these materials can impart to reverse electrodialysis setups has to do with the fact that the barriers are nanoscopically thin with high pore densities, and have been set up in such a way as to prevent ionic leak currents; you couldn't simply disperse them in a CS or AS hydrogel section of equal dimensions and expect improved performance). I believe the authors understand this, but a reader new to this concept may not.

4. The authors claim to be powering a "smartwatch," but the watch shown looks like the most basic LCD time display imaginable – far from "smart." This should be amended to something like "digital wristwatch."

Avenues of possible future inquiry:

1. To me, one of the most interesting things to come out of this work is the systematic exploration of the dependency of voltage decay kinetics on the ratio of the gel lengths – lengthening the HS and LS gels while keeping the CS and AS gels the same length produces a substantial "plateau" region of up to 40-50 hours in which the open-circuit voltage stays approximately stable before decaying exponentially; this comes at the cost of added resistance (Fig. 4d-g). The original work on this general hydrogel arrangement (ref. 12's table S2 and Extended Data Fig. 5) shows that the

majority of the electrical resistance of such systems comes from the LS gel. I accordingly wonder if some of the benefits of this "plateau" region could be captured without adding to the resistance if only the HS gel were lengthened without lengthening the LS gel.

2. The most conspicuous omission from this paper, to me, was any discussion of recharging the system via the application of potential/current to the terminal electrodes (electrodialysis, rather than reverse electrodialysis), which has been demonstrated before in these systems (e.g. ref. 12's Supplementary Information section S3 and Extended Data Fig. 3). The absence of any discussion of recharging is particularly notable given that the authors show applications involving the integration of this power scheme into wearables. Accordingly, I think the authors could consider characterizing the recharge performance of the hydrogel fibers they are able to create in future manuscripts.

Best,
Thomas Schroeder
North Carolina State University

Reviewer #4 (Remarks to the Author):

In this novel work, the authors present an automated extrusion-based fabrication method for multimaterial hydrogel salinity-gradient batteries which produce voltages of up to 208V, withstand strains of approximately 150%, and perform under a wide range of operating temperatures. The methods presented are a significant improvement upon their previous high-impact work [13]: using an elegant setup, a well-ordered and sensible search of the relevant printing parameters is first performed, before the resulting material properties are thoroughly investigated and a broad number of applications are presented.

I believe that this work is suitable for publication in Nature Communications, and I have only minor comments to be addressed:

1. Could the authors comment on the mechanical longevity of the battery with respect to its 84h half-life from Table 1 – does the hydrogel dry out to the point of non-usability following water losses such as that mentioned in line 175?
2. Line 14: "...to efficiently fabricate biomimetic ionic hydrogel batteries with a maximum stretchability of 200%." I think this phrasing is misleading, given that Figure 3a's tests of the battery unit reach only 150%*.
3. Could the authors comment more on the apparent drift of voltage occurring throughout Figure 3h?
4. Many of the figures would be easier to interpret if values for the scale bars were added to the figure itself: for example, in Figure 2, three separate values (2, 5 & 10mm) are used, which can only be discovered by reading through a long caption.
5. Line 56: "...thin papers were separately penetrated the four kinds of hydrogel precursor solutions..." I cannot understand what is being described in this sentence.
6. General spelling and grammar checks would be beneficial throughout, including the occasional spelling mistakes in the figure labels (such as "Reservoirs" in Figure 1a & "Porpotion" in Figure 4e).

* All other mentions of stretchability throughout the manuscript make clear that 200% applies to only one of the filaments proposed, whereas that is not the case here.

Reviewer #5 (Remarks to the Author):

The authors devised a homemade apparatus for fabricating a series of ion concentration gradient batteries inspired by the electric eel. They optimized the recipe of the hydrogels and meticulously evaluated the mechanical properties of each component. Subsequently, biomimetic batteries were efficiently fabricated in series or parallel with seamless interface connections. This study developed the printing technology for hydrogel-based biomimetic batteries with detailed parameters. The referee would like to recommend publishing this work in Nature Communications after addressing the following concerns.

1. Each battery unit has 137 mV of open circuit voltage that originates from the electrochemical potential difference of HS and LS hydrogels. Do the authors attempt to enhance the open circuit voltage (OCV) by regulating the ion concentration in both electrode hydrogels? Furthermore, could the authors provide the relationship between ion concentration and OCV using the Nernst equation, if feasible
2. The viscosity-shear rate curves depicted in Figure 2a demonstrate the optimal composition achieved with 1.3% HEC and 0.05% PEO. What are the potential implications if the concentration of HEC exceeds 1.3%? Additionally, in Figure 2c, a notable disparity in G'/G'' is observed within the high-frequency range of 10^2 to 10 Hz. How might this discrepancy affect the fabrication of the batteries?
3. Lines 216-223, "The CS hydrogel showed the highest elongation of $200.05\% \pm 14.38\%$, and that of HS, LS, and AS hydrogel was $77.57\% \pm 9.64\%$, $154.13\% \pm 31.56\%$, and $103.21\% \pm 41.62\%$, respectively.....to form a seamless ionic hydrogel battery unit, it exhibited 221 compromised flexibility with an elongation of $137.47 \pm 26.97\%$". Is there a "barrel effect" from HS that has the lowest elongation? LS has the second-highest elongation of 154%, but it was fractured first in the strain test, why?
4. Lines 235-236, "the internal resistance of the ionic hydrogel battery unit increased in proportion to the elongation ratio, resulting in a 96.6% rise in the internal resistance at an ϵl of 100%." Generally, the resistivity of a material remains constant under specific conditions. The resistance is proportional to the sample's length and inversely proportional to the cross-sectional area, how do the author think about only 96.6% rise in the internal resistance at an ϵl of 100%?
5. The open circuit voltages (OCVs) decline due to self-discharge resulting from spontaneous ion diffusion. Are there any potential strategies that could be employed to address this challenge in the future?
6. Line 439, "which enabled to print g the ionic hydrogel battery", what is the meaning of the "g"?
7. Please provide the power of the LED bulbs to allow readers to evaluate the performance of the printed batteries.
8. Before submission, please thoroughly review the reference information. Approximately 20% of the references appear to be incomplete, such as refs. 2, 5, 10, 11, 16, 17, 27, 40.

Response to Reviewer:

We wish to thank the reviewers for their constructive comments, which will improve the clarity and quality of our paper. The reviewers' comments are included in *italic* and our responses follow in **red**.

In the revised manuscript, all the changes are highlighted in **red**.

Response to Reviewer #1:

Reviewer #1 (Remarks to the Author): The authors reported a novel consecutive multimaterial printing strategy to fabricate flexible battery filaments for applications. The design can assemble the ionic hydrogel batteries in 30 mins with a high voltage of 208 V. in addition, this strategy can be applied on various flexible power device applications. The result is good, and the discussion is very systematically. Therefore, this paper can be accepted on Nature communications after addressing the following questions.

1. Detailed discussions for the anti-freezing mechanism of the hydrogel electrolyte should be provided, which can not be just attribute to the seamless interface.

We thank the reviewer for pointing this out. As the reviewer suggested, we have systematically conducted an anti-freezing experiment on the as-printed soft ionic hydrogel power source and provided a detailed discussion of the anti-freezing mechanism in the revised manuscript. The consecutively-printed soft ionic hydrogel power source exhibits a remarkable discharge performance even at extremely low temperatures (-20 °C), which is mainly attributed to the incorporation of anti-freezing compositions within the hydrogel matrix, as well as the seamless interface among different hydrogel portions. As shown in **Supplementary Figure 17, four types of**

hydrogel portions exhibited different freezing tolerances at $-20\text{ }^{\circ}\text{C}$. The HS and LS hydrogels maintained their unfrozen and transparent state even after being stored at $-20\text{ }^{\circ}\text{C}$ for 24 hours as shown in **Supplementary Figure 17b**. This phenomenon is attributed to the incorporation of lithium chloride (*Chemical Engineering Journal* 419 (2021) 129478; *Journal of Materials Chemistry C* 11 (2023) 10573-10583; *International Journal of Biological Macromolecules* 227 (2023) 462-471), and glycerol (*Journal of Materials Science* 56 (2021) 18697-18709) inside the hydrogel matrix, which have been proven a feasible strategy to prepare anti-freezing hydrogels. Although the transparency of the CS and AS hydrogels without the addition of salts decreased due to the partial freezing of free water, they still retained the flexibility and can stand twisting deformation without fracturing even after being stored at $-20\text{ }^{\circ}\text{C}$ for 24 hours (**Supplementary Figure 17c**). According to previous literature, the partially-frozen CS and AS hydrogels contain “weakly bound water” and “unfrozen water” within the hydrogel network (*Materials Horizons* 8 (2021) 351-369). As a result, the consecutively-printed hydrogel power source at $-20\text{ }^{\circ}\text{C}$ still maintained unfrozen water and seamless interfaces, which can provide a continuous conductive pathway of ion transportation for voltage output.

Supplementary Figure 17. Anti-freezing tests for the four types of hydrogel materials. (a) The transparency of four types of hydrogel materials stored at room temperature and (b) at $-20\text{ }^{\circ}\text{C}$ for 24 hours. (c) The flexibility of four types of hydrogel materials after storage at $-20\text{ }^{\circ}\text{C}$ for 24 hours.

To make it clear to the readers, we added relevant discussion in the main text, and incorporated the corresponding results in the supplementary information (**Supplementary Figure 17**). The revision can be read as “Moreover, the consecutively-printed ionic hydrogel power source can also work under a wide temperature range (-20 °C -90 °C). Typically under the conditions of -20 °C, the HS and LS hydrogels exhibited exceptional anti-freezing properties due to the presence of lithium chloride and/or glycerol, whereas the CS and AS hydrogels experienced partial freezing with “weakly bound water” and “unfrozen water” inside the hydrogel matrix (**Supplementary Figure 17**) (*Materials Horizons* 8 (2021) 351-369). The presence of unfrozen water along with the seamless interface of the as-printed ionic hydrogel filament enables the stable connection and the continuous ionic transportation for stable voltage output even at a low temperature of -20 °C.”

Please find changes in red on **Page 29, Line 474-479**.

2. The water retention performance of the ionic hydrogel should be characterized at different environment such as humidity and temperature.

As the reviewer suggested, the water retention performance of four types of hydrogel materials and the consecutively-printed ionic hydrogel power source unit with a fixed length of 48.0 mm was investigated under different temperature and humidity conditions (25 °C and 45% RH, 60 °C and 45% RH, 25 °C and 90% RH) for 12 hours. The photographs of the packed and unpacked ionic hydrogel power source units incubated at different conditions are shown in **Supplementary Figure 6a**. All the packed ionic hydrogel power source units can maintain their original shapes with little change in appearance. The initial water content of the samples was set as 100%. The water content of the hydrogels at different time points was further calculated by the following equation:

$$\text{The water content} = \frac{m_n - m_{dry}}{(m_0 - m_{dry})} \times 100\%$$

Where, m_0 is the initial weight of ionic hydrogel power source units, m_{dry} is the dry weight of ionic hydrogel power source units after being stored at 90 °C for 12 hours, and m_t is the weight of ionic hydrogel power source units incubated at different environments for 12 hours.

As shown in **Supplementary Figure 6b**, the environmental factors showed a significant impact on the water contents of the unpacked ionic hydrogel power source units, which can be calculated as $17.82\% \pm 2.08\%$ at 25 °C and 45% RH, $3.30\% \pm 0.43\%$ at 60 °C and 45% RH, and $125.48\% \pm 5.05\%$ at 25 °C and 90% RH. In contrast, the water content maintained at a relatively stable level for the ionic hydrogel power source units packed inside flexible films, which facilitate the stable electric discharge across a broad range of temperatures and humidity. Similar phenomena were also observed for the four types of hydrogel filaments as shown in **Supplementary Figure 6c-f**. For the unpacked hydrogel filaments, the water loss exceeded 87% under a relatively small humidity. As the humidity increased to 90%, the hydrogel filaments can absorb water from the air and the water content increased to over 120%. In contrast, the water contents of the packed hydrogel filaments exhibited little fluctuation when exposed to different environmental conditions for 12 hours. These results indicated that the packed ionic hydrogel power sources fabricated by the presented consecutive printing strategy had excellent environmental adaptability.

To make this clear to the readers, we added more information on the water retention performance of the ionic hydrogel power sources in the revised main text, which can be read as “The consecutively-printed IHPS was packed with flexible substrate (polyethylene film with a thickness of 200 μm) aiming to avoid water loss and simultaneously ensuring electrically isolation between the printed ionic hydrogel filaments to prevent short-circuiting even under the large deformation

conditions. The film-packed ionic hydrogel power sources maintained a stable water content across a broad range of temperature and humidity (**Supplementary Figure 6**), which is crucial for the long-term stable electric discharge.” The related results were added in **Supplementary Figure 6**.

Please find the changes in red on **Page 11, Line 202-207**.

Supplementary Figure 6. Water retention tests of packed and unpacked consecutive ionic hydrogel power source units and four types of hydrogel materials. (a) Photographs and (b) water content of packed and unpacked ionic hydrogel power source unit stored at 25 °C and 45% RH, 60 °C and 45%

RH, 25 °C and 90% RH for 0 and 12 hours. Water content of packed and unpacked (c)HS, (d) LS, (e) AS and (f) CS hydrogel filament stored at 25 °C and 45% RH, 60 °C and 45% RH, 25 °C and 90% RH for 12 hours.

3. For mechanical strength characterizations, the cycling compressive stress–strain curves and cyclic tensile loading–unloading test for the hydrogel electrolyte can be provided.

According to the reviewer’s suggestion, cyclic compressive and tensile tests were conducted on the hydrogel samples by using an electromechanical universal testing machine (103A, Wance, China).

For the cyclic tensile loading-unloading tests, the four types of hydrogel filaments with an initial length of 48 mm were stretched to a maximum strain of 50% for 50 cycles by using a 10 N load cell at a crosshead speed of 4 mm s⁻¹. **Supplementary Figure 8a-d** show the stress-strain curves as well as the images of the hydrogel filaments before and after cyclic tensile loading-unloading tests. All kinds of hydrogel filaments can recover their original shapes without any sign of rupture and the corresponding stress-strain curves showed good overlap among different loading-unloading cycles.

The tensile properties of the consecutively-printed ionic hydrogel power source units were further investigated by gradually increasing the tensile strains from 20% to 100%, and the corresponding stress-strain profiles are shown in **Supplementary Figure 8e**. It can be seen that the stress-strain curves at a lower strain can gradually and smoothly transform to the neighbor loop at a higher strain.

The final cumulative profile showed negligible difference to the stress-strain curves of the ionic hydrogel power source units directly stretched to a strain of 100% (**Supplementary Figure 8f**).

Little change was observed when the ionic hydrogel power source unit was periodically subjected

to a strain of 100% for 50 cycles. These results indicated that the consecutively-printed ionic hydrogel power source units have excellent elastic recovery and resilience capability.

Supplementary Figure 8. Cyclic tensile tests of the four hydrogel electrolyte materials and the consecutive ionic hydrogel power source unit. Cyclic tensile stress-strain curves of (a) HS hydrogel, (b) LS hydrogel, (c) AS hydrogel, and (d) CS hydrogel under a strain of 50% for 50 cycles for 50 cycles by using a 10 N load cell at a crosshead speed of 4 mm s^{-1} . Insets show images of the four types of hydrogel samples before and after 50 stretching cycles under a strain of 50%. (e) Tensile

stress-strain curves of ionic hydrogel power source unit at a strain of 20%, 40%, 60%, 80% and 100%, respectively. (f) Cyclic tensile stress-strain curves of ionic hydrogel power source unit under a maximum strain of 100% for 50 cycles. Insets show images of the ionic hydrogel power source unit before and after 50 stretching cycles.

The cyclic compression tests were conducted on the four types of cylindrical hydrogel samples with a diameter of 13.0 mm and a height of 6.5 mm. The samples were subject to a strain of 50% for 50 cycles by using a 50 N load cell at a crosshead speed of 1 mm s^{-1} . Unlike the cyclic tensile tests, a hysteresis loop can be clearly observed in each compressive loading-unloading cycle as shown in **Supplementary Figure 9**. During cyclic compression tests, each kind of hydrogel sample exhibited a similar reproducible stress-strain curve pattern. This feature may endow the ionic hydrogel power source with good electric discharge stability under repeated compressive conditions.

Supplementary Figure 9. Cyclic compressive tests of the four hydrogel electrolyte materials.

Cyclic compressive stress-strain curves of (a) HS hydrogel, (b) LS hydrogel, (c) AS hydrogel, and (d) CS hydrogel. Insets show images of the four hydrogel samples before and after 50 compression cycles.

To make this clear to the readers, we added the results in **Supplementary Figure 8-9** and **Supplementary Video 1**. The methods for cyclic mechanical testing were added in the revised main text. The description and discussion on these results were added in the main text, which can be read as “Besides, cyclic tensile and compression tests were conducted on the hydrogel samples. The tensile stress-strain curve of the ionic hydrogel power source unit showed a high degree of overlap over strains ranging from 20% to 100% as well as under a constant strain of 100% among different loading-unloading cycles (**Supplementary Figure 8**). Additionally, the ionic hydrogel power source unit can maintain its original shape without any sign of rupture under a strain of 100% in tension for 50 cycles, indicating excellent elastic recovery and resilience capability. During cyclic compression tests, each kind of hydrogel sample exhibited similar reproducible stress-strain curve patterns and can restore its original shape without fracture (**Supplementary Figure 9**). The excellent mechanical robustness of these ionic hydrogels was mainly attributed to their abundant reversible ionic and hydrogen bonds within the hydrogel matrix networks(*European Polymer Journal* 168 (2022) 111099), making them promising for potential applications in the fields of soft robotics and flexible electronics.” Please find the changes in red on **Page 14, Line 242-252**.

4. FTIR, Raman or other spectra characterizations can be provided to illustrate the interactions in the hydrogel for better performance design.

According to the reviewer's suggestion, FTIR spectra of different hydrogel components were characterized by using a Fourier transform infrared spectrometer (Nicolet iS10, Thermo Fisher, US) operating at a resolution of 1 cm^{-1} . The FTIR spectra were recorded in the mid-infrared (MIR) range of $400\text{--}4000\text{ cm}^{-1}$ (**Supplementary Figure 2a**). The FTIR spectra of HS and LS possessed a similar profile since the main difference between their formulations was the concentration of lithium chloride, showing no infrared absorption. The characteristic peaks in the FTIR spectra of HS and LS hydrogels were mainly observed at $3200\text{--}3530\text{ cm}^{-1}$, $2950\text{--}2880\text{ cm}^{-1}$, 1652 and 1602 cm^{-1} , attributed to --N--H stretching, --C--H stretching, C=O , and N--H in the --CONH and the angular deformations of $\text{--CH}_2\text{--}$, respectively. These observations were consistent with that of polyacrylamide (*Advanced Functional Materials* (2018) 673-680). The FTIR spectra of AS hydrogel also showed a similar profile to that of polyacrylamide, as the characteristic peaks of the anion-selective reagents overlapped with those of polyacrylamide at similar wavelengths. The broad band at $2450\text{--}3500\text{ cm}^{-1}$ of CS hydrogel was attributed to the --N--H stretching, --C--H stretching of polyacrylamide, and --O--H stretching of the cation-selective reagent, respectively.

The Raman spectra of different hydrogel components were recorded using a Raman Spectroscopy system (LabRAM HR Evolution, HORIBA, Japan) with a 532 nm laser from 100 to 4000 cm^{-1} at a resolution of 0.2 cm^{-1} . As shown in **Supplementary Figure 2b**, the band at 2930 cm^{-1} was attributed to the C--H stretching of polyacrylamide, and the broad peaks at $3000\text{--}3700\text{ cm}^{-1}$ were due to the amide --N--H , and the water O--H . The vibration bands corresponding to hydrogen bonds are centered at 3248 and 3445 cm^{-1} , which are assigned to strongly hydrogen-bonded --O--H stretching and weakly hydrogen-bonded --O--H stretching in the hydrogel components, respectively (*Solid State Ionics* 145 (2001) 79-84). It can be seen that a relatively large number of hydrogen

bonds were formed in the HS hydrogel, due to the addition of a large amount of LiCl (*Advanced Science* 10 (2023) 2303922). The spectra from 100 – 1800 cm^{-1} are dominated by the internal vibrations of the polymeric acrylamide chains.

Supplementary Figure 2. (a) FTIR spectra of HS, LS, AS, and CS hydrogels characterized by using a Fourier transform infrared spectrometer (Nicolet iS10, Thermo Fisher, US) operating at a resolution of 1 cm^{-1} and (b) Raman spectra of HS, LS, AS, and CS hydrogels recorded using a Raman Spectroscopy system (LabRAM HR Evolution, HORIBA, Japan) with a 532 nm laser from 100 to 4000 cm^{-1} at a resolution of 0.2 cm^{-1} .

To make this clear to the readers, we have added the results in the **Supplementary Figure 2**. The description and discussion of these results were added in the main text, which can be read as “The four types of hydrogels with optimized contents of rheology modifiers were further characterized by FTIR and Raman spectroscopy. The peaks observed at 3200–3530 cm^{-1} in the FTIR spectra of four types of hydrogels belong to the N–H stretching of polyacrylamide, while the peaks around 1652 cm^{-1} are the C=O characteristic absorption peak of it as shown in **Supplementary Figure 2**. The broad band at 2450–3500 cm^{-1} of CS hydrogel was attributed to the N–H stretching, C–H stretching of polyacrylamide, and O–H stretching of the cation-selective reagent, respectively. A relatively large number of hydrogen bonds were found in the Raman spectra of HS hydrogel due to

the addition of a large amount of LiCl (*Advanced Science* 10 (2023) 2303922), resulting in good anti-freezing property.” Please find the changes in red on **Page 8, Line 131-138**.

5. *Can the length of each section in the hydrogel battery influence the performance?*

This is a good question. The electrical properties of ionic hydrogel power sources were highly determined by the amount of ions inside the HS and LS hydrogel components depended on the length of each portion, which offers a new opportunity to regulate battery discharge properties by changing the switching frequency. **Figure 4d** shows the consecutively-printed ionic hydrogel power sources with 10 units in series when the switching frequency for CS and AS components was constantly fixed at 1/4 Hz while that for the HS and LS components was changed into 1/8 Hz, 1/12 Hz, 1/16 Hz, and 1/20 Hz (**Supplementary Video 4**), respectively. It can be clearly seen that the CS and AS portions maintained the same length of 12.04 ± 0.15 mm while the length of the HS and LS portions gradually increased from 24.16 ± 0.21 to 60.09 ± 0.16 mm. Consequently, the V_{oc} remained relatively stable, while the I_{sc} remarkably decreased from 17.79 ± 2.36 μ A to 8.93 ± 1.59 μ A due to prolonged unit length (**Figure 4e**). The current-voltage curves of the consecutively-printed hydrogel power sources with varying HS and LS proportions exhibited similar and linear profiles with different slopes (**Figure 4f**), confirming the scalability and tunability of their electrical properties. Moreover, the discharge rate of the ionic hydrogel power source was significantly decelerated by the increase in the length of HS and LS portions. As shown in **Figure 4g**, the V_{oc} of ionic hydrogel power sources with an HS:CS:LS:AS portion ratio of 8:4:8:4 exhibited an exponential decline over time, whereas that of the power sources with a portion ratio of 12:4:12:4 initially maintained a stable V_{oc} (>90% of original V_{oc}) for 14.95 ± 0.54 h before experiencing an

exponential decrease. Further increasing the HS and LS proportion to 16 and 20 resulted in a stable V_{oc} of the consecutively-printed ionic hydrogel power sources for more than 27.36 ± 0.74 h and 48.71 ± 0.68 h. Similarly, the time for the loss of 50% of the original V_{oc} was also significantly prolonged from 14.35 ± 0.26 h for ionic hydrogel power sources with a proportion of 8:4:8:4 to 84.24 ± 0.63 h (proportion of 20:4:20:4). These phenomena were primarily attributed to the augmented ionic concentrations and elongated ionic migration pathways of both HS and LS portions. Additionally, it was found that simply elongating the ion transfer pathway by increasing the length of both CS and AS portions can effectively decrease the discharge rate while maintaining initial V_{oc} magnitude (**Supplementary Figure 13**), demonstrating a high degree of tunability in their power dissipation profiles.

Figure 4. Regulation of the switching frequency to modulate the proportion and obtain desired electrical characteristics of the consecutively-printed ionic hydrogel power sources. (d) Photographs, (e) V_{oc} , I_{sc} , (f) current-voltage curves, and (g) power dissipation curves of the consecutively-printed ionic hydrogel power sources containing 10 units in series with the HS:CS:LS:AS proportion varying from 8:4:8:4, 12:4:12:4, 16:4:16:4 to 20:4:20:4.

Reviewer #2 (Remarks to the Author): I thoroughly enjoyed reading this well-arranged paper. Hydrogels are become very hot topic and show their broad applications in various fields, including batteries and sensors. The manuscript underscores the distinct advantage of direct printing of multi-materials over the conventional method of transferring prefabricated hydrogels into molds. To realize the consecutive printing process, four types of ionic hydrogel inks were developed, resulting in seamlessly bonded battery filaments. The filaments exhibit stable voltage outputs during repeatable stretching, culminating in an impressive voltage of 208 V. As this work details the printing of hydrogel and its impact on salinity power, I believe it is fit to Nat comm., pending minor revision. The following issues need to be addressed:

(1) The demonstrated batteries appear to be one-time use. Is there potential for enhancing durability by implementing a sustainable salt gradient, akin to the strategy proposed in Energy Environ. Sci. 2017, 10, 1923 “Solar-driven simultaneous steam production and electricity generation from salinity”

We would like to thank the reviewer for the valuable suggestion. As the reviewer mentioned, the as-printed ionic hydrogel power sources in the current study are one-time use since the salt gradient between HS and LS hydrogel components would gradually decrease, which limited their applications as long-term power sources with stable voltage output. However, the ionic hydrogel power source units can gradually retrieve their original voltage output of *c.a.* 137 mV within 30 minutes by using a direct current (DC) power generator as shown in **Supplementary Figure 18**, which endows the as-printed ionic hydrogel power source recharging property for repetitive use. Moreover, as the reviewer mentioned, it is crucial to establish regenerated ion concentration gradients for constructing ionic hydrogel power sources with constant voltage output. The

previously developed solar hybrid system (*Environ. Sci.* 2017, 10, 1923 “Solar-driven simultaneous steam production and electricity generation from salinity”) can generate steam and concentrated seawater at the same time, which might provide an efficient way to regenerate ion gradients. Therefore, the integration of the solar hybrid system and this RED-based ionic hydrogel power source into one system holds promise for the development of sustainable RED devices with consistent voltage output for long-term utilization.

To make this clear to the readers, we have added more discussion and the literature in the main text to illustrate potential strategies for enhancing the durability of the present ionic hydrogel power sources, which can be read as "One of the major challenges for the present ionic hydrogel power sources is the gradually decreased voltage output, as the ion gradients decrease between the HS and LS hydrogels during discharging. Future efforts to address this challenge could be the integration of efficient solar desalination processes that can regenerate ion-concentration gradients (*Energy & Environmental Science* 10 (2017) 1923-1927; *Energy Storage Materials* 37 (2021) 556-566; *Desalination* 535 (2022) 115824) with the RED-based power sources, thereby forming closed-loop-systems capable of establishing sustainable salinity power generators for long-term use."

Please find changes in red on **Page 30, Line 499-504**.

(2) I noticed that the current-voltage curves were obtained by connecting hydrogel batteries using resistors. Could you furnish a typical IV curve by scanning voltage with an electrochemical workstation ?

As the reviewer suggested, the IV curve of the consecutively-printed ionic hydrogel power source unit was further measured based on a conventional two-electrode method by using an

electrochemical workstation (MetrohmAutolab PGSTAT302N, Switzerland). The applied voltage ranges from 0 to 1 V. As shown in **Supplementary Figure 4**, the IV curve of the ionic hydrogel power source unit showed a basically linear relationship between applied voltage and current, exhibiting typical ohmic-like ionic current properties (*Chemical Science* 8 (2017) 890-913; *Advanced Intelligent Systems* 1 (2019) 1900073). This result is consistent with the results obtained from the load-switching tests. To make this clear to the readers, we have added the relevant results in the main text, which can be read as “Besides, the IV curve of the ionic hydrogel power source unit showed a basically linear relationship between applied voltage and current (**Supplementary Figure 4**), exhibiting typical ohmic-like ionic current properties”. Please find changes in red on **Page 11, Line 191-193**.

Supplementary Figure 4. The IV curve of consecutively-printed ionic hydrogel power source unit using an electrochemical workstation (MetrohmAutolab PGSTAT302N, Switzerland) with a conventional two-electrode method with the applied voltage ranging from 0 to 1 V.

(3) *The claim of printed batteries achieving a maximum voltage of 208 V and a half-life of 84 h requires evidences to bolster credibility.*

We have added the testing video to demonstrate the maximum voltage and the power dissipation process of the ionic hydrogel power source in **Supplementary Video 6** and **Video 9**. Please find the Video in the supplementary information.

(4) It remains unclear whether the printed hydrogel batteries are sealed. Will dehydration happen during measurement?

We thank the reviewer to point this out. In this study, the consecutively-printed ionic hydrogel power source unit was packed with a flexible substrate (polyethylene film with a thickness of 200 μm) to avoid dehydration during the measurement process. We have tested the water contents of the printed hydrogel filaments with and without packing (**Supplementary Figure 6**). It was found that the water contents of the unpacked ionic hydrogel power source units experienced significant changes, which can be calculated as $17.82\% \pm 2.08\%$ at 25 °C and 45% RH, $3.30\% \pm 0.43\%$ at 60 °C and 45% RH, and $125.48\% \pm 5.05\%$ at 25 °C and 90% RH. In contrast, the water content maintained at a relatively stable level for the ionic hydrogel power source units packed inside flexible films, which facilitate the stable electric discharge across a broad range of temperature and humidity.

To make this clear to the readers, we have added more information on the pack method and water retention performance of the ionic hydrogel power sources in the revised main text, and the related results were added in the supplementary information of **Supplementary Figure 6**, which can be read as “The consecutively-printed ionic hydrogel power source unit was packed with flexible substrate (polyethylene film with a thickness of 200 μm) to isolate the hydrogel filament with one another, which avoided mutual short-circuiting and any potential water loss during discharging. The

film-packed IHPS maintained a stable water content across a broad range of temperature and humidity (**Supplementary Figure 6**), which is crucial for the long-term stable electric discharge.”

Please find changes in red on **Page 11, Line 202-207**.

Supplementary Figure 6. Water retention tests of packed and unpacked consecutively-printed ionic hydrogel power source units. (a) Photographs and (b) water content of packed and unpacked ionic hydrogel power source unit stored at 25 °C and 45% RH, 60 °C and 45% RH, 25 °C and 90% RH for 0 and 12 h.

Reviewer #3 (Remarks to the Author): In this work, the authors present a new fabrication approach and overall morphology for a hydrogel-based ion-gradient electrical power generation/storage (“battery”) scheme first reported by Schroeder et al in 2017 (ref. 12). They have engineered the system such that they can print continuous multi-component fibers that each consist of the repeating series arrangement of four distinct hydrogels necessary to produce the intended power transduction effect. The resulting fibers are reasonably durable and flexible; a single repeating unit can be stretched substantially (to a strain of 150%, corresponding to a stretch ratio of 2.5 times the original length) and twisted and knotted in various arrangements without breaking. The authors also explored printing in a Peano curve morphology, which allows substantially more deformation. They explored in detail the effects of modulating the length of the various hydrogel units (framed in the paper as changing the switching frequency of their printing setup) on the voltage and current generation capacity and discharge kinetics of the system, producing some interesting results. They finally demonstrated that they can exploit the flexibility, fiber geometry, and electrical modularity of this scheme to power a wearable device (an LCD watch).

The authors have done a very thorough job of engineering this system for production, which involves complex multimaterial printing and curing on a flexible substrate that is continuously moved in a roll-to-roll system; this sort of demonstration is of definite value to this field, as it represents a step toward practical implementation at scale. The data in the work supports the claims of the paper; I have not found flaws in the analysis or data interpretation. The methods are generally presented clearly and in detail (though I have a minor comment about this below). I accordingly recommend this work for publication after a few very minor revisions are made for clarity’s sake, and I congratulate the authors on what they’ve been able to accomplish here.

I will now detail the minor revisions, and also suggest some possible avenues of further inquiry that the authors could pursue in future work, since they seem well-set-up to do so.

The minor revisions:

1. The authors must include the full compositions of the prepolymer solutions in their Methods section, rather than saying that they were prepared “according to the previously established protocols” and referring to reference #13. This is for two reasons: first, it’s a basic courtesy to save the reader from having to look up another paper, but secondly because the paper referenced is not open-access like this one is; readers may have access only to this manuscript.

We appreciate the reviewer for pointing this out. According to the reviewer’s suggestion, we have added a detailed composition of hydrogel materials in the Methods section, which can be read as “All the ionic hydrogel precursors were prepared by dissolving corresponding reagents into ultra-pure water. HS precursor: 6.0 mol L⁻¹ lithium chloride, 0.054 mol L⁻¹ bis, 4.38 mol L⁻¹ acrylamide, 0.007 mol L⁻¹ photoinitiator. LS precursor: 0.015 mol L⁻¹ lithium chloride, 0.062 mol L⁻¹ bis, 4.10 mol L⁻¹ acrylamide, 3.4 mol L⁻¹ glycerol, 0.015 mol L⁻¹ photoinitiator. AS precursor: 2.0 mol L⁻¹ (3-acrylamidopropyl)trimethyl ammonium chloride, 0.034 mol L⁻¹ bis, 2.75 mol L⁻¹ acrylamide, 0.012 mol L⁻¹ photoinitiator. CS precursor: 2.0 mol L⁻¹ 2-acrylamido-2-methylpropane sulfonic acid, 0.055 mol L⁻¹ bis, 1.90 mol L⁻¹ acrylamide, 0.012 mol L⁻¹ photoinitiator. To enhance the stability of CS ink, hydroquinone was introduced into the CS precursor as a polymerization inhibitor at a concentration of 0.15 mg/mL. To enable efficient switching of four types of hydrogel inks during consecutive printing, the resulting inks were formulated by incorporating HEC rheology modifiers at concentrations of 1.30%, 1.52%, 1.35%, and 1.82% into the HS, LS, CS, and AS precursors, respectively, while maintaining a constant PEO concentration of 0.05%.”

2. I am assuming (and I think I can see visually) that the printed fibers in Figures 3h-i and 6e-g are electrically isolated from one another using clear plastic wrap in order to prevent different parts of the hydrogel assembly from short-circuiting with each other via the flow of ions. This could be easily missed and should be both explicitly mentioned and explained in the text of the paper.

We thank the reviewer for pointing this out. We have revised the figures and added the relevant information in the revised manuscript, which can be read as “The consecutively-printed ionic hydrogel power source was packed with flexible substrate (polyethylene film with a thickness of 200 μm) aiming to avoid water loss and simultaneously ensuring electrical isolation between the printed ionic hydrogel filaments to prevent short-circuiting even under the large deformation conditions.” Please find changes in red on **Page 11, Line 202-205**.

3. I have an issue with how the authors have worded lines 475-477 of the discussion section of the paper: “Additionally, incorporating conductive nanomaterials such as MXene, carbon nanotubes into the hydrogel matrix with designed microstructures such as aligned microchannels and nonporous structures can also enhance the ion flux with reduced internal resistance.” First, I believe the authors meant to write “nanoporous” instead of “nonporous”. More generally, all the referenced papers (refs 28-35) describe innovations in membrane materials in order to achieve high selectivity, high overall permeability, and low resistance, as the authors say, and they could in fact be useful in a scheme like the authors have described. However, I think the authors need to make clear that the inclusion of the types of nanomaterials and membrane structures described in these papers would **replace** the function of the cation and anion selective gels, rather than enhancing

their functionality as dopants in the gel. (A lot of the improvement in resistance these materials can impart to reverse electrodialysis setups has to do with the fact that the barriers are nanoscopically thin with high pore densities, and have been set up in such a way as to prevent ionic leak currents; you couldn't simply disperse them in a CS or AS hydrogel section of equal dimensions and expect improved performance). I believe the authors understand this, but a reader new to this concept may not.

We appreciate the reviewer for pointing this out. We agree with the reviewer that the as-mentioned references can achieve high selectivity and high overall permeability at a nanoscale thickness, which might be not suitable for the present all-hydrogel power source. However, previous literature has also shown that the millimeter-scale ion-selective hydrogels membrane with aligned microchannels in parallel to the ion transportation direction exhibited a faster ion diffusion rate than those with channels perpendicular to the ion transportation direction. (*Nano Energy* 102 (2022)) Similar strategies to construct 3D ion transport channels in the ion-selective membranes have proven to be feasible to enhance the ion flux. (*Advanced Functional Materials* 29 (2019) 1900326; *Nano Energy* 97 (2022) 107170) Therefore, it is believed that the constructing aligned and interconnected channels inside the ion-selective hydrogels in parallel to the ion transportation direction might enhance the ion diffusion rate and further increase the output power of the present ionic hydrogel power sources.

To make this clear to the readers, we have replaced the previously used reference paper and revised the related description. They can be read as “The low output power of ionic hydrogel power sources was mainly caused by the lower throughput of ion transport flux, which was attributed to the limited selectivity and the high internal resistance of the hydrogel matrix. (*Lab on a Chip* 16 (2016) 700-

708; *Adv Mater* (2021) 2101757) It has been proven a feasible way to improve the power output of the ionic hydrogel power sources by reducing their internal resistance by incorporating conductive materials such as PEDOT:PSS(*Energy Storage Materials* 49 (2022) 348-359) and MXene(*Chemical Society Reviews* 49 (2020) 7229-7251) into the hydrogel matrix as well as shortening the ion conductive pathways by further increasing the switching frequency. Apart from this, there is a need to develop novel ion-selective hydrogels that possess high permeability, exceptional ion selectivity, and low internal resistance (*Advanced Functional Materials* (2021) 2009586). For example, constructing aligned microchannels inside the ion-selective hydrogels in parallel to the ion transportation direction might enhance the ion diffusion rate by decreasing ion diffusion pathway. (*Advanced Functional Materials* 29 (2019) 1900326; *Nano Energy* 102 (2022))” Please find changes in red on **Page 29, Line 490-499** in the revised manuscript.

4. *The authors claim to be powering a “smartwatch,” but the watch shown looks like the most basic LCD time display imaginable – far from “smart.” This should be amended to something like “digital wristwatch.”*

We thank the reviewer for pointing this out. We have replaced the “smartwatch” with “digital wristwatch” in the revised manuscript.

Avenues of possible future inquiry:

1. *To me, one of the most interesting things to come out of this work is the systematic exploration of the dependency of voltage decay kinetics on the ratio of the gel lengths – lengthening the HS and LS gels while keeping the CS and AS gels the same length produces a substantial “plateau” region*

of up to 40-50 hours in which the open-circuit voltage stays approximately stable before decaying exponentially; this comes at the cost of added resistance (Fig. 4d-g). The original work on this general hydrogel arrangement (ref. 12's table S2 and Extended Data Fig. 5) shows that the majority of the electrical resistance of such systems comes from the LS gel. I accordingly wonder if some of the benefits of this "plateau" region could be captured without adding to the resistance if only the HS gel were lengthened without lengthening the LS gel.

We thank the reviewer for the valuable suggestion. As the reviewer suggested, we have consecutively printed ionic hydrogel power sources with a unit proportion of 20:4:4:4 to investigate the short-circuit current and power dissipation profile. The length of HS hydrogel component is 5 times higher than the LS hydrogel component. The I_{sc} of the unit was measured at $19.36 \pm 4.25 \mu\text{A}$, which was higher than that of the unit with a proportion of 20:4:20:4 ($8.93 \pm 1.59 \mu\text{A}$). This mainly caused by the reduced internal resistance due to the shortened LS hydrogel component. **Supplementary Figure 12** shows the power dissipation profile of the ionic hydrogel power source with 10 units in series (proportion of 20:4:20:4). The half-life of the ionic hydrogel power source with a proportion of 20:4:4:4 was significantly extended by increasing the HS portion compared with that with a proportion of 4:4:4:4, which is consistent with our finding that the half-life can be extended with elongated ionic migration pathways. However, purely increasing HS portion in the unit did not show an obvious "plateau" region in the power dissipation profile. A clear plateau region was found when the length of HS and LS portions simultaneously increased. This phenomenon may be attributed to the lower LS proportion, resulting in a smaller volume and a shorter ion migration pathway.

Supplementary Figure 12. Power dissipation profile of ionic hydrogel power source with 10 units in series with a proportion of 20:4:4:4 compared with that of 4:4:4:4 and 20:4:20:4.

To make this clear to the readers, we have added the results in the revised main text, and the related results were added in **Supplementary Figure 12**, which can be read as “Notably, the I_{sc} of the ionic hydrogel power source with a proportion of 20:4:4:4 was improved to $19.36 \pm 4.25 \mu\text{A}$ by purely lengthening the HS portion in the unit, which was higher than that of the unit with a proportion of 20:4:20:4 due to the reduced internal resistance of the shortened LS hydrogel component. The half-life of the ionic hydrogel power source with a proportion of 20:4:4:4 was significantly extended by increasing the HS portion compared with that with a proportion of 4:4:4:4, which is consistent with our finding that the half-life can be extended with elongated ionic migration pathways (**Supplementary Figure 12**). However, purely increasing HS portion in the unit did not show an obvious “plateau” region in the power dissipation profile as that of 20:4:20:4, which may be attributed to the lower LS proportion, resulting in a smaller volume and a shorter ion migration pathway.” Please find changes in red on **Page 19, Line 332-341**.

2.The most conspicuous omission from this paper, to me, was any discussion of recharging the

system via the application of potential/current to the terminal electrodes (electrodialysis, rather than reverse electrodialysis), which has been demonstrated before in these systems (e.g. ref. 12's Supplementary Information section S3 and Extended Data Fig. 3). The absence of any discussion of recharging is particularly notable given that the authors show applications involving the integration of this power scheme into wearables. Accordingly, I think the authors could consider characterizing the recharge performance of the hydrogel fibers they are able to create in future manuscripts.

Best,

Thomas Schroeder

North Carolina State University

We thank the reviewer for pointing this out. As the reviewer suggested, we performed recharging experiments on the consecutively-printed ionic hydrogel power source unit. An external DC power generator (RIGOL DP712, China) with a voltage output of 10 V was used to drive the ionic flow from LS hydrogels to HS hydrogels. It was found that the ionic hydrogel power source unit was able to retrieve a voltage output of 136.56 ± 10.34 mV after one discharge and recharge cycle for 30 minutes, which was close to the initial voltage of *c.a.* 137 mV.

Supplementary Figure 18. The voltage output of the ionic hydrogel power sources basically recovered after being recharged with a voltage of 10 V for 30 minutes by a DC voltage generator (RIGOL DP712, China).

Reviewer #4 (Remarks to the Author): In this novel work, the authors present an automated extrusion-based fabrication method for multimaterial hydrogel salinity-gradient batteries which produce voltages of up to 208V, withstand strains of approximately 150%, and perform under a wide range of operating temperatures. The methods presented are a significant improvement upon their previous high-impact work [13]: using an elegant setup, a well-ordered and sensible search of the relevant printing parameters is first performed, before the resulting material properties are thoroughly investigated and a broad number of applications are presented.

*I believe that this work is suitable for publication in Nature Communications, and I have only **minor** comments to be addressed:*

1. Could the authors comment on the mechanical longevity of the battery with respect to its 84h half-life from Table 1 – does the hydrogel dry out to the point of non-usability following water losses such as that mentioned in line 175?

We thank the reviewer for pointing this out. In this study, the consecutively-printed ionic hydrogel power source was packed with flexible substrate (polyethylene film with a thickness of 200 μm) aiming to avoid water loss and simultaneously ensuring electrical isolation between the printed ionic hydrogel filaments to prevent short-circuiting even under the large deformation conditions. It was found that the water contents of the unpacked ionic hydrogel power source units experienced significant changes, which can be calculated as $17.82\% \pm 2.08\%$ at 25 °C and 45% RH, $3.30\% \pm 0.43\%$ at 60 °C and 45% RH, and $125.48\% \pm 5.05\%$ at 25 °C and 90% RH. In contrast, the water content maintained at a relatively stable level for the ionic hydrogel power source units packed inside flexible films, which facilitate the stable electric discharge across a broad range of temperature and humidity over a long period.

To make this clear to the readers, we have added more information on the pack method and water retention performance of the ionic hydrogel power sources in the revised main text, and the related results were added in **Supplementary Figure 6**, which can be read as “The consecutively-printed ionic hydrogel power source unit was packed with flexible substrate (polyethylene film with a thickness of 200 μm) to isolate the hydrogel filament with one another, which avoided mutual short-circuiting due to ionic mobility and water loss during discharging. The film-packed ionic hydrogel power sources maintained a stable water content over a wide range of temperature and humidity conditions (**Supplementary Figure 6**), which was crucial for the long-term stable electric discharge.” Please find changes in red on **Page 11, Line 202-207**.

Supplementary Figure 6. Water retention tests of packed and unpacked consecutive ionic hydrogel power source units. (a) Photographs and (b) water content of packed and unpacked ionic hydrogel power source unit stored at 25 °C and 45% RH, 60 °C and 45% RH, 25 °C and 90% RH for 0 and 12 hours.

The ionic hydrogel power source (proportion of 20:4:20:4) with 10 units in series was packed with flexible film to avoid water loss. Therefore, the resulting ionic hydrogel power sources following a discharge period of 100 hours can maintain excellent foldability, rollability, and twistability without any signs of damage, as demonstrated in **Supplementary Figure 19**. This observation highlights

the remarkable flexibility longevity of the as-printed ionic hydrogel power source even after undergoing a long-term discharge duration.

Supplementary Figure 19. The photographs of ionic hydrogel power source (proportion of 20:4:20:4) with 10 units in series after discharging for 100 hours under (a) folded, (b) rolled and (c) twisted conditions and recovered to flat state.

2. Line 14: “...to efficiently fabricate biomimetic ionic hydrogel batteries with a maximum stretchability of 200%.” I think this phrasing is misleading, given that Figure 3a’s tests of the battery unit reach only 150%*.

We thank the reviewer for pointing this out. We have revised the description as “...to efficiently fabricate biomimetic ionic hydrogel power sources with a maximum stretchability of ~137%” in the revised main text.

3. Could the authors comment more on the apparent drift of voltage occurring throughout Figure 3h?

We thank the reviewer for pointing this out. The open-circuit voltage profile of the consecutively-printed ionic hydrogel power source unit was recorded in **Figure 3h**, which underwent manually folded, twisted, and knotted deformations. To measure the voltage output, two silver electrodes were

embedded in the HS hydrogel at both ends of the ionic hydrogel power source unit. It was found that the voltage of the ionic hydrogel power source unit under manual deformation conditions slightly fluctuated within the range of 9 mV (less than 7% of the initial voltage), which might mainly result from the unstable contact between the relatively-rigid electrodes and soft hydrogel (**Figure 3h**).

To make this clear to the readers, we added the discussion in the revised main text, which can be read as “Besides high stretchability, the consecutively-printed ionic hydrogel power source unit also exhibited high flexibility and robustness under various deformation conditions including manual folding, twisting, and knotting, as evidenced by the stable V_{oc} with a fluctuation of less than 7% (**Figure 3h**). The voltage drift during manual deformation mainly results from the unstable contact between the relatively-rigid electrodes and soft hydrogel. Afterward, the ionic hydrogel power source unit can restore its initial shape and voltage output after deformation.” Please find changes in red on **Page 15, Line 276-281**.

4. Many of the figures would be easier to interpret if values for the scale bars were added to the figure itself: for example, in Figure 2, three separate values (2, 5 & 10mm) are used, which can only be discovered by reading through a long caption.

We thank the reviewer for pointing this out. All figures have been carefully checked and revised to clearly show the scale bar and relevant information. Please find changes in the revised manuscript.

5. Line 56: “...thin papers were separately penetrated the four kinds of hydrogel precursor solutions...” I cannot understand what is being described in this sentence.

We revised the sentence to make it clear to readers, which can be read as “To improve the current output, ionic hydrogel power sources with shortened ionic transport pathway were constructed by infiltrating four hydrogel precursor solutions into thin paper films (hundreds of microns thick) and then stacking them sequentially.” Please find changes in red on **Page 4, Line 57-59**.

6. General spelling and grammar checks would be beneficial throughout, including the occasional spelling mistakes in the figure labels (such as “Reserviors” in Figure 1a & “Porpotion” in Figure 4e).

** All other mentions of stretchability throughout the manuscript make clear that 200% applies to only one of the filaments proposed, whereas that is not the case here.*

We thank the reviewer for pointing this out. We have carefully revised all the figures and grammar in the revised manuscript. The stretchability of the consecutively-printed ionic hydrogel power source is described according to its specific case in the revised main text.

Reviewer #5 (Remarks to the Author): The authors devised a homemade apparatus for fabricating a series of ion concentration gradient batteries inspired by the electric eel. They optimized the recipe of the hydrogels and meticulously evaluated the mechanical properties of each component. Subsequently, biomimetic batteries were efficiently fabricated in series or parallel with seamless interface connections. This study developed the printing technology for hydrogel-based biomimetic batteries with detailed parameters. The referee would like to recommend publishing this work in Nature Communications after addressing the following concerns.

1. Each battery unit has 137 mV of open circuit voltage that originates from the electrochemical potential difference of HS and LS hydrogels. Do the authors attempt to enhance the open circuit voltage (OCV) by regulating the ion concentration in both electrode hydrogels? Furthermore, could the authors provide the relationship between ion concentration and OCV using the Nernst equation, if feasible.

We thank the reviewer for pointing this out. This study mainly focused on the development of the consecutive multimaterial printing technique and the potential applications of ionic hydrogel power sources. The concentration and formulation of the four types of hydrogel precursors were chosen according to previous literature (*Adv Mater* (2021) 2101757). The effect of different types of cations and anions and varied concentrations on the electrical performance of the resultant power source has been previously investigated. According to the reviewer's suggestion, the voltage of the ionic hydrogel power source unit with different concentrations of 2.0 mol L⁻¹, 4.0 mol L⁻¹, and 6.0 mol L⁻¹ in HS was tested as shown in **Supplementary Figure 3**. It can be seen that the voltage of the ionic hydrogel power source unit increased as the concentration of LiCl in the HS hydrogel was improved. To make this clear to the readers, the related results were added in the supplementary information

of **Supplementary Figure 3**, and the relevant description was added in the revised main text which can be read as “The V_{oc} of a single IHPS unit was 137.89 ± 18.10 mV, which increased with a larger concentration gradient between HS and LS (**Supplementary Figure 3**).” Please find changes in red on **Page 10, Line 179-180** in the revised manuscript.

Supplementary Figure 3. The voltage of ionic hydrogel power source unit with a concentration of LiCl of 2.0 mol L^{-1} , 4.0 mol L^{-1} , and 6.0 mol L^{-1} in the HS hydrogel.

As the reviewer mentioned, the open-circuit voltage across the ion-selective hydrogel of an ionic hydrogel power source unit can be theoretically related to the ion concentration through the utilization of the Nernst-Planck equation and Goldman-Hodgkin-Katz current equation (*Nature* 552 (2017) 214) as follows.

$$V_{oc} = \frac{RT}{F} \ln \left(\frac{P_{c, Li^+} [Li^+]_{ex} + P_{c, Cl^-} [Cl^-]_{in}}{P_{c, Li^+} [Li^+]_{in} + P_{c, Cl^-} [Cl^-]_{ex}} \right)$$

V_{oc} (V) is the open-circuit voltage across the cation-selective hydrogel, R ($\text{J mol}^{-1} \text{K}^{-1}$) is the gas constant, T (K) is the temperature, F (C mol^{-1}) is Faraday’s constant, P_{c, Li^+} and P_{c, Cl^-} (m s^{-1}) is the permeability of Li^+ and Cl^- through cation-selective hydrogel respectively, $[i^+]_{in}$ and $[i^+]_{ex}$ (mol m^{-3}) are the concentrations of ion i inside and outside the membrane.

The V_{oc} of the ionic hydrogel power source unit was determined by the cumulative potential differences across the anion- and cation-selective membrane. The V_{oc} of the ionic hydrogel power

source increased with a higher ratio of a larger concentration gradient or ion permeabilities between HS and LS, which was in good agreement with our findings as illustrated in **Supplementary Figure 3**.

2. The viscosity-shear rate curves depicted in Figure 2a demonstrate the optimal composition achieved with 1.3% HEC and 0.05% PEO. What are the potential implications if the concentration of HEC exceeds 1.3%? Additionally, in Figure 2c, a notable disparity in G'/G'' is observed within the high-frequency range of 10^2 to 10 Hz. How might this discrepancy affect the fabrication of the batteries?

We thank the reviewer for pointing this out. As shown in **Supplementary Figure 1**, the printed HS filament was prone to spread out on the substrate at a smaller HEC content of 1.2%. In contrast, the filamentary shape can be well maintained when the HEC content increased to 1.3%. However, further increasing the non-conductive rheology modifier HEC concentration resulted in increased internal resistance of the hydrogel matrix, which hindered the power output of the resultant ionic hydrogel power source. As a result, the concentration of HEC in the HS ink was optimized at 1.3% to minimize internal resistance while ensuring good printability. A similar approach was also employed to optimize the concentration of HEC in the initial LS, CS, and AS precursor solutions for achieving consistent rheological properties during consecutive multimaterial printing of ionic hydrogel filaments with uniform dimensions. To make this clear to the readers, we have added related descriptions in the revised manuscript. They can be read as “However, further increasing the concentration of non-conductive rheology modifier HEC resulted in increased internal resistance of the hydrogel matrix, which will hinder the power output of the resultant ionic hydrogel power source.

As a result, the concentration of HEC in the HS ink was optimized at 1.3% to minimize internal resistance while ensuring good printability.” Please find changes in red on **Page 7, Line 116-120** in the revised manuscript.

Supplementary Figure 1. The photograph of printed HS hydrogel filaments with different concentrations of HEC ranging from 1.0% to 1.3% (w/v %) at an interval of 0.1% and 0.05% PEO at the air pressure of 120 kPa.

The storage modulus and loss modulus of the optimized four types of ionic hydrogel power source inks are shown as a function of the angular frequency in **Figure 2c**. As the reviewer mentioned, G'' significantly surpassed G' at high angular frequencies. This observation indicates that the developed ionic hydrogel power source inks behave as viscoelastic liquids under high angular frequencies, which is a favorable rheological property for extrusion-based 3D printing (*Journal of Manufacturing Processes* 35 (2018) 526-537; *3D Printing and Additive Manufacturing* 10 (2021) 816-827; *Adv Mater* 34 (2022) 2108855). In this case, the viscoelastic liquid property of the ink at a high shear rate might be beneficial for multimaterial printing, which requires rapid extrusion and frequent switching of the developed four inks inside the multimaterial printhead. In contrast, G'' merely slightly exceeded G' in the low-frequency range below 10 rad/s. This property further

ensured the shape fidelity of the hydrogel filaments after extruded from the printhead.

To make this clear to the readers, we have added related descriptions in the revised manuscript. They can be read as “The storage modulus (G') and loss modulus (G'') of the optimized four types of ionic hydrogel power source inks are shown as a function of the angular frequency in **Figure 2c**. The G'' significantly surpassed G' at high angular frequencies, which is a favorable rheological property for extrusion-based 3D printing. In contrast, G'' merely slightly exceeded G' in the low-frequency range below 10 rad/s, indicating that the inks exhibited a semi-gel state under a low shear rate. These properties are very important for the rapid extrusion and frequent switching of the developed four inks inside the multimaterial printhead, and simultaneously maintain the shape fidelity of the hydrogel filaments after extruded from the printhead.” Please find the changes in red on **Page 8, Line 124-131**.

3. Lines 216-223, “The CS hydrogel showed the highest elongation of $200.05\% \pm 14.38\%$, and that of HS, LS, and AS hydrogel was $77.57\% \pm 9.64\%$, $154.13\% \pm 31.56\%$, and $103.21\% \pm 41.62\%$, respectively.....to form a seamless ionic hydrogel battery unit, it exhibited 221 compromised flexibility with an elongation of $137.47 \pm 26.97\%$”. Is there a “barrel effect” from HS that has the lowest elongation? LS has the second-highest elongation of 154%, but it was fractured first in the strain test, why?

In this study, the “barrel effect” for the tensile properties of the consecutive ionic hydrogel power source unit resulted from the LS hydrogel due to its lowest break strength as shown in **Figure 3a**.

In a multicomponent system, the portion with the lower break strength fractured first during stretching. The break strength of HS, LS, CS, and AS hydrogels was 9.32 ± 0.28 kPa, 6.27 ± 0.32

kPa, 13.04 ± 0.37 kPa and 19.76 ± 0.47 kPa, respectively. Therefore, the LS hydrogel fractured first in the tensile test.

To make this clear to the readers, we have added more descriptions in the revised main text, which can be read as “Moreover, when the strain further increased to $137\% \pm 26.97\%$, the LS hydrogel was fractured first since it has lowest break strength of 6.27 ± 0.32 kPa in the ionic hydrogel power source unit (Supplementary Figure 7).” Please find changes in red on Page 14, Line 239-241.

Figure 3. (a) The stress-strain curves of the consecutively-printed ionic hydrogel power source units and single-component printed hydrogel filaments.

4. Lines 235-236, “the internal resistance of the ionic hydrogel battery unit increased in proportion to the elongation ratio, resulting in a 96.6% rise in the internal resistance at an el of 100%.”

Generally, the resistivity of a material remains constant under specific conditions. The resistance is proportional to the sample’s length and inversely proportional to the cross-sectional area, how do the author think about only 96.6% rise in the internal resistance at an el of 100%?

We appreciate the reviewer to point this out. We agree with the reviewer that in an ideal electroconductive resistor, resistance is directly proportional to the length of the sample and inversely proportional to its cross-sectional area. However, it should be noted that our system

comprises a hydrogel-based ionic power source consisting of four distinct components, namely HS, CS, LS, and AS hydrogel. As such, it belongs to ionic conductive structures characterized by inherent differences in ion concentration. The concentration gradient of LiCl between LS and HS acts as the driving force for active ion transportation, while CS and AS govern the rate of free ion transport. Therefore, the internal resistance of the current ionic hydrogel power sources may primarily depend on the efficiency of materials selectively conducting ions. Additionally, previous studies have also demonstrated that stretching increases the inhomogeneity of the hydrogel matrix and reduces its cross-sectional area, which slows down free ion conduction and increases resistance. However, stable contacts remained among ions within the hydrogel, resulting in a lower change in ionic resistance compared to ideal resistance. (*NPG Asia Materials* 14 (2022) 11; *Cellulose* 29 (2022) 5725-5743; *Chemical Engineering Journal* 472 (2023) 144849). Hence, the underlying mechanisms of strain-induced resistance in ionic hydrogel power sources remain elusive, and our previous statement regarding the linear proportion relationship between the internal resistance of one unit and its elongation ratio is imprecise. To avoid any potential misinterpretation, we have modified the relevant description as follows: “The relative resistance change exhibits a step increase corresponding to the increase in strain (**Figure 3d**), which increased to 94.04% at an elongation of 100%. The strain-induced resistance change of the IHPS unit was different compared with an ideal electroconductive resistor. This phenomenon might be caused by the inhomogeneous elongation of four types of hydrogels with different initial ion concentration and ion-selective efficiency in one unit, which still necessitates a comprehensive investigation in the future.” Please find changes in red on **Page 15, Line 258-263**.

5. *The open circuit voltages (OCVs) decline due to self-discharge resulting from spontaneous ion diffusion. Are there any potential strategies that could be employed to address this challenge in the future?*

This is a good question. The voltage of the RED-based ionic hydrogel power source decreased with the decreasing ion gradients between the HS and LS hydrogels. To construct ionic hydrogel power sources with constant voltage, it is crucial to establish regenerated ion concentration gradients. The previously developed solar hybrid desalination system can generate steam and concentrated seawater at the same time, which might provide an efficient way to regenerate ion gradients. (*Energy & Environmental Science* 10 (2017) 1923-1927; *Energy Storage Materials* 37 (2021) 556-566; *Desalination* 535 (2022) 115824) Therefore, the integration of the solar hybrid system and this RED-based ionic hydrogel power source into one system holds promise for the development of sustainable RED devices with consistent voltage output for long-term utilization.

To make this clear to the readers, we have added more discussion in the main text to illustrate potential strategies for enhancing the durability of the present ionic hydrogel power sources, which can be read as "One of the major challenges for the present ionic hydrogel power sources is the gradually decreased voltage output, as the ion gradients decrease between the HS and LS hydrogels during discharging. Future efforts to address this challenge could be the integration of efficient solar desalination processes that can regenerate ion-concentration gradients (*Energy & Environmental Science* 10 (2017) 1923-1927; *Energy Storage Materials* 37 (2021) 556-566; *Desalination* 535 (2022) 115824) with the RED-based power sources, thereby forming closed-loop-systems capable of establishing sustainable salinity power generators for long-term use."

Please find changes in red on **Page 30, Line 499-504.**

6. Line 439, “which enabled to print g the ionic hydrogel battery”, what is the meaning of the “g”?

We thank the reviewer for pointing this out. We have removed the letter in the revised manuscript.

7. Please provide the power of the LED bulbs to allow readers to evaluate the performance of the printed batteries.

We thank the reviewer for pointing this out. The nominal power of LED bulb is 1.3 mW. We have added the relevant information in the revised manuscript.

8. Before submission, please thoroughly review the reference information. Approximately 20% of the references appear to be incomplete, such as refs. 2, 5, 10, 11, 16, 17, 27, 40.

We thank the reviewer to point this out. We have revised the reference information in the revised manuscript.

REVIEWER COMMENTS

Reviewer #1 (Remarks to the Author):

This paper can be accepted for publication without change.

Reviewer #2 (Remarks to the Author):

The authors have addressed most of my concerns. My second question is about the IV curve of the generator. The curve should not cross the zero point. The authors could refer to Figure 6 in J. Am. Chem. Soc. 2014, 136, 12265.

Reviewer #3 (Remarks to the Author):

The authors have implemented all the changes I asked for, and have even gone above and beyond my expectations by testing both of the avenues of future inquiry I'd laid out for inclusion in this paper. I have no remaining issues with this work, so I strongly recommend it for publication in Nature Communications. I again commend the authors for bringing this field closer to practical implementation, which is gratifying to see.

Best,
Thomas Schroeder
North Carolina State University

Reviewer #4 (Remarks to the Author):

My minor concerns have all been satisfactorily addressed by the authors. I believe that the revised manuscript is suitable for publication.

Reviewer #5 (Remarks to the Author):

The authors addressed the comments with additional data carefully in the revised manuscript. It could be accepted at the current state.

Response to Reviewer:

We wish to thank the reviewer for their constructive comments, which will improve the clarity and quality of our paper. The reviewers' comments are included in *italic* and our responses follow in **red**.

In the revised manuscript, all the changes are highlighted in **red**.

Response to Reviewer #2:

Reviewer #2 (Remarks to the Author): The authors have addressed most of my concerns. My second question is about the IV curve of the generator. The curve should not cross the zero point. The authors could refer to Figure 6 in J. Am. Chem. Soc. 2014, 136, 12265.

We would like to thank the reviewer for pointing this out. In the first round of revision, we followed the reviewer's suggestion to measure the IV curve of the printed ionic hydrogel power source by using an electrochemical workstation (MetrohmAutolab PGSTAT302N, Switzerland). The obtained result was added in Supplementary Figure 4. We enlarged the IV curve and found that the curve did not cross the zero point in the high-magnification image.

According to the new suggestion from the reviewer, we have carefully read the referenced literature (*Journal of the American Chemical Society* 136 (2014) 12265-12272) and characterized the IV properties of the consecutive ionic hydrogel power source with a single unit or 5 units in series by using a Keithley electrometer (6517b). The sweeping voltage from -1 V to +1 V was applied onto the working electrode with a step voltage of 0.02 V. For each kind of ionic hydrogel power source, three samples were tested and the results were shown in **Supplementary Figure 4**. It can be seen that the open-circuit voltage and short-circuit current of the ionic hydrogel power source unit were

134.81 \pm 6.62 mV and 25.98 \pm 2.31 μ A, respectively. When the unit number increased to 5, the corresponding open-circuit voltage linearly increased to 670.43 \pm 26.49 mV while the short-circuit current remained relatively stable at 26.01 \pm 1.32 μ A. For the same kind of power source, the *IV* curves showed little change among the three samples. These results were consistent with the *IV* curves obtained from the load-switching technique as shown in **Figure 2m**.

To make this clear to the readers, we have added the discussion on the relevant results in the main text, which can be read as “The *IV* curve of the ionic hydrogel power source was further characterized by using the linear sweep voltammetry. The V_{oc} and I_{sc} of ionic hydrogel power source unit read from the intercepts on the voltage and current axes were 134.81 \pm 6.62 mV and 25.98 \pm 2.31 μ A (**Supplementary Figure 4**), (*Journal of the American Chemical Society* 136 (2014) 12265-12272) respectively, which was consistent with the results of the normalized current and voltage curves. When the unit number increased to 5, the corresponding V_{oc} linearly increased to 670.43 \pm 26.49 mV while the I_{sc} remained relatively stable at 26.01 \pm 1.32 μ A.” Please find the change on

Page 10, Line 188-193.

Supplementary Figure 4. The *IV* curves of consecutively-printed ionic hydrogel power source with a single unit and 5 units in series by using an electrometer (Keithley 6517b, America) with the

applied voltage ranging from -1 V to 1 V with a step voltage of 0.02 V.

REVIEWERS' COMMENTS

Reviewer #2 (Remarks to the Author):

The authors have addressed my concerns. Consequently, I recommend publishing this manuscript as is.